# Insights into the dynamic control of breathing revealed through cell-type-specific responses to substance P

**Nathan A Baertsch[1]\*, Jan-Marino Ramirez[1,2]**

[1]Center for Integrative Brain Research, Seattle Children's Research Institute, Seattle, United States; [2]Department of Neurological Surgery, University of Washington School of Medicine, Seattle, United States

**Abstract** The rhythm generating network for breathing must continuously adjust to changing metabolic and behavioral demands. Here, we examined network-based mechanisms in the mouse preBötzinger complex using substance P, a potent excitatory modulator of breathing frequency and stability, as a tool to dissect network properties that underlie dynamic breathing. We find that substance P does not alter the balance of excitation and inhibition during breaths or the duration of the resulting refractory period. Instead, mechanisms of recurrent excitation between breaths are enhanced such that the rate that excitation percolates through the network is increased. We propose a conceptual framework in which three distinct phases of inspiration, the burst phase, refractory phase, and percolation phase, can be differentially modulated to control breathing dynamics and stability. Unraveling mechanisms that support this dynamic control may improve our understanding of nervous system disorders that destabilize breathing, many of which involve changes in brainstem neuromodulatory systems.

**\*For correspondence:**
nathan.baertsch@seattlechildrens.org

## Introduction

Rhythmicity is ubiquitous in the brain, important for many high-order functions such as consciousness, attention, perception, and memory (*Başar and Düzgün, 2016*; *Başar and Güntekin, 2008*; *Colgin, 2016*; *Hanslmayr et al., 2016*; *Kiehn, 2016*; *Neske, 2015*; *Palva and Palva, 2018*; *Paton and Buonomano, 2018*), as well as vital rhythmic motor behaviors including chewing, locomotion, and breathing (*Grillner and El Manira, 2015*; *Kiehn, 2016*; *Narayanan and DiLeone, 2017*; *Ramirez and Baertsch, 2018a*; *Wyart, 2018*; *Nakamura et al., 2004*). Rhythms generated by the brain are diverse, as are the underlying rhythm generating network- and cellular-level mechanisms (*Paton and Buonomano, 2018*). Indeed, rhythmicity can occur on time scales ranging from milliseconds to days (*Golombek et al., 2014*). However, most neuronal rhythms must be flexible, able to increase or decrease the rate of oscillation or the degree of synchronization to match changes in physiological or cognitive demands (*Brittain et al., 2014*; *Ramirez and Baertsch, 2018b*). Thus, understanding principles that allow dynamic control of rhythm generating networks may provide important insights into the regulation of diverse brain functions.

For half a century, investigations of the networks and cellular mechanisms that generate breathing have provided valuable, generalizable insights into the origins and control of neural rhythmicity (*Del Negro et al., 2018*; *Feldman and Kam, 2015*; *Ramirez and Baertsch, 2018b*; *Wyman, 1977*; *Cohen, 1981*; *Ezure, 1990*; *Long and Duffin, 1986*; *Milsom, 1991*). Inspiration, the dominant phase of breathing in mammals (*Jenkin and Milsom, 2014*), is generated by a spatially dynamic network located bilaterally along the ventrolateral medulla (*Baertsch et al., 2019*). A region that is both necessary and sufficient for inspiration, the pre-Bötzinger Complex (preBötC), is autorhythmic and forms the core of this network (*Smith et al., 1991*; *Tan et al., 2008*; *Vann et al., 2018*). Like many

rhythmic networks, a critical characteristic of the preBötC is that the frequency of its output is dynamic - the rate of breathing changes during for example sleep/wake states, exercise, environmental challenges, and orofacial behaviors such as feeding and vocalization (*Moore et al., 2013*; *Ramirez et al., 2016*). Excitatory and inhibitory inputs from other brain regions, as well as neuromodulation, can potently facilitate or depress the frequency of breathing (*Doi and Ramirez, 2008*; *Zuperku et al., 2017*). Yet, elucidating how these influences alter the network- and cellular-level rhythm generating mechanisms within the preBötC remains a challenge (*Dick et al., 2018*).

Glutamatergic synaptic interactions among preBötC interneurons allow this sparsely connected network (*Carroll and Ramirez, 2013*; *Schwab et al., 2010*) to periodically synchronize and are therefore obligatory for rhythmogenesis (*Ge and Feldman, 1998*). However, if left unrestrained, feed-forward excitation in the network leads to hyper synchronization during inspiratory bursts, which subsequently causes a prolonged period of reduced network excitability (*Baertsch et al., 2018*; *Kottick and Del Negro, 2015*). This refractory phase delays the onset of the next inspiratory burst and, as a result, the frequency of breathing becomes very slow. Inhibitory interactions within the preBötC are critical for limiting synchronization during bursts (*Harris et al., 2017*) and reducing the subsequent refractoriness of the network (*Baertsch et al., 2018*). Indeed, roughly 40% of inspiratory preBötC neurons are inhibitory (GABAergic and/or glycinergic) (*Oke et al., 2018*; *Winter et al., 2009*; *Morgado-Valle et al., 2010*) and by regulating the excitability of glutamatergic neurons during inspiratory bursts, these neurons play an important role in controlling breathing frequency.

However, controlling the inspiratory burst itself is not the only mechanism that regulates the inspiratory rhythm. During the time between bursts, referred to as the inter-burst interval (IBI), recurrent excitatory synaptic connections (*Del Negro et al., 2018*) and ectopic bursting of individual preBötC neurons (*Ramirez et al., 2004*) are thought to give rise to a gradual increase in network excitability that drives the onset of the next burst. In this model, spontaneous spiking activity in a small subset of excitatory preBötC neurons begins to percolate stochastically through the network, gradually recruiting more spiking activity among interconnected excitatory neurons (*Kam et al., 2013b*). During this percolation phase, activation of membrane voltage- and calcium-dependent conductances in an increasing number of neurons causes the excitation to become exponential, culminating in an inspiratory burst (*Del Negro et al., 2010*; *Ramirez et al., 2016*). The gradual increase in spiking activity during this phase, or 'pre-inspiratory ramp', is thought to be primarily mediated by a subset of glutamatergic preBötC interneurons that have enhanced excitability. Derived from V0-lineage precursors, these neurons express the transcription factor developing brain homeobox one protein (Dbx1) during development (referred to here as 'Dbx1 neurons') (*Bouvier et al., 2010*; *Gray et al., 2010*; *Picardo et al., 2013*; *Wu et al., 2017*). How this process of recurrent excitation may contribute to the dynamic regulation of inspiratory frequency is not well understood.

Here, we examine network- and cellular-level changes in the inspiratory rhythm generator that underlie dynamic frequency responses to the excitatory neuromodulator substance P (SP). A member of the tachykinin neuropeptide family, SP is a key mediator of many physiological and neurobiological processes (*Mantyh, 2002*). For breathing, SP regulates the stability of the respiratory rhythm (*Ben-Mabrouk and Tryba, 2010*; *Yeh et al., 2017*) as well as respiratory responses to hypoxia (*Chen et al., 1990*; *Ptak et al., 2002*). The endogenous receptor for SP, neurokinin one receptor (NK$_1$R), is expressed on only ~5–7% of CNS neurons (*Mantyh, 2002*), but is enriched in the preBötC (*Gray et al., 1999*; *Schwarzacher et al., 2011*). SP binding to NK$_1$R causes excitation of preBötC neurons through coupling with voltage-independent cation channels (*Hayes and Del Negro, 2007*; *Ptak et al., 2009*), including sodium leak channel, non-selective (Nalcn). Disruption of this ion channel causes pathological respiratory instability (*Yeh et al., 2017*). SP also promotes robust facilitation of inspiratory frequency (*Gray et al., 1999*; *Peña and Ramirez, 2004b*). Therefore, SP is an ideal tool to explore the changes in network activity that regulate breathing stability and underlie the dynamic control of breathing frequency.

By combining electrophysiological, optogenetic, and pharmacological techniques, we find that SP differentially influences the refractory and recurrent excitation phases of the inspiratory rhythm through cell-type-specific effects. We conclude that phase-specific and differential modulation of excitatory and inhibitory network interactions is a key mechanism that allows the frequency of this vital rhythmogenic network to be dynamically controlled.

## Results

### SP has differential effects on the refractory and percolation phases of the preBötC rhythm

To explore how the neuromodulator SP increases inspiratory frequency at the network level, integrated preBötC population activity was recorded in horizontal brainstem slices (*Anderson et al., 2016*; *Baertsch et al., 2019*). from Dbx1$^{CreERT2}$;Rosa26$^{ChR2-EYFP}$ neonatal mice during bath application of 0.5–1.0 μM SP (n = 6). Horizontal slices that contain the ventral respiratory column were chosen over transverse slices that attempt to isolate the preBötC because inspiratory neurons rostral of the preBötC may be important for the dynamic regulation of the refractory period and breathing frequency (*Baertsch et al., 2019*). In response to SP, inspiratory burst frequency increased from 0.23 ± 0.02 Hz to 0.31 ± 0.03 Hz (p=0.003) during steady state SP (>~3 min post bath application) similar to previous observations in transverse slice preparations (*Peña and Ramirez, 2004b*). To determine whether the refractory period is modulated by SP, brief light pulses (200 ms, 0.5 mW/mm$^2$) were delivered randomly during the inspiratory cycle at baseline and in SP. In each condition, the probability of light-evoking a burst in the contralateral preBötC was quantified as a function of elapsed time from the preceding spontaneous population burst and compared to the cumulative distribution of spontaneous inter-burst intervals (IBIs). A representative experiment is shown in *Figure 1A,B* and the average data are shown in *Figure 1C*. Evoked bursts were rare if a light pulse occurred immediately following a spontaneous population burst. However, the probability of a light-evoked burst increased with elapsed time until ~2 s following a spontaneous burst when bursts could be evoked with nearly every light pulse (*Figure 1A,B*). The end of this ~2 s refractory period coincided with a large increase in the number of spontaneous IBIs (*Figure 1B,C*), indicating that this period of reduced preBötC excitability precludes both light-evoked and spontaneous preBötC burst generation, thereby preventing very short IBIs and fast inspiratory rhythms (*Baertsch et al., 2018*). However, despite a frequency increase of 31.9 ± 5.6%, the refractory period was not altered by SP (non-linear regression analysis; p>0.05). In SP, IBIs remained limited by the refractory period, but spontaneous bursts occurred more quickly and more consistently following the end of the refractory period. As a result, the average IBI became shorter (4.5 ± 0.4 s to 3.5 ± 0.5 s; paired t-test; p<0.0002) and less variable (SD of 1.3 ± 0.1 to 0.84 ± 0.1; p<0.0139) in SP (*Figure 1C*). Together these data suggest that SP increases the frequency and regularity of the inspiratory rhythm through differential modulation of two inspiratory phases: The refractory phase remains unchanged, while the duration of the recurrent excitation or percolation phase, which promotes the onset of the subsequent burst, is reduced.

### Inspiratory spiking patterns of excitatory and inhibitory neurons in the preBötC

Next, we explored the spiking patterns of individual excitatory and inhibitory preBötC neurons to identify mechanisms that may underlie differential modulation of the refractory and percolation phases in the preBötC network. Inspiratory spiking activity was recorded from n = 29 preBötC neurons in whole-cell (WC) or cell-attached (CA) configuration (*Figure 2*). Example cell-attached recordings are shown in *Figure 2—figure supplement 1* and confirmed our results in whole-cell configuration. Membrane potentials of neurons recorded in whole-cell configuration are shown in *Figure 2—figure supplement 2*. Horizontal slices from Vglut2$^{Cre}$;Rosa26$^{ChR2-EYFP}$ and Vgat$^{Cre}$; Rosa26$^{ChR2-EYFP}$ mice were used so that recorded neurons could be identified as excitatory or inhibitory based on depolarizing responses to light. In Vglut2$^{Cre}$;Rosa26$^{ChR2-EYFP}$ slices, light pulses were delivered following blockade of synaptic transmission to determine the intrinsic response of the neuron (*Baertsch et al., 2018*; *Baertsch et al., 2019*). Recorded neurons were also fluorescently labeled using patch pipets containing AlexaFluor568 to mark their anatomical locations (*Figure 2A*). There was considerable variability among inspiratory neurons (maximal spiking activity during inspiration), with respect to spike frequency, burst duration, and burst shape; and for any given neuron there was considerable burst-to-burst stochasticity (*Carroll and Ramirez, 2013*; *Carroll et al., 2013*). However, excitatory neurons could be clearly grouped based on the presence or absence of spiking during the IBI that typically increases in frequency, or 'ramps', before the subsequent inspiratory burst – often referred to as 'pre-inspiratory (pre-I)' activity. In contrast, we did not identify any inspiratory

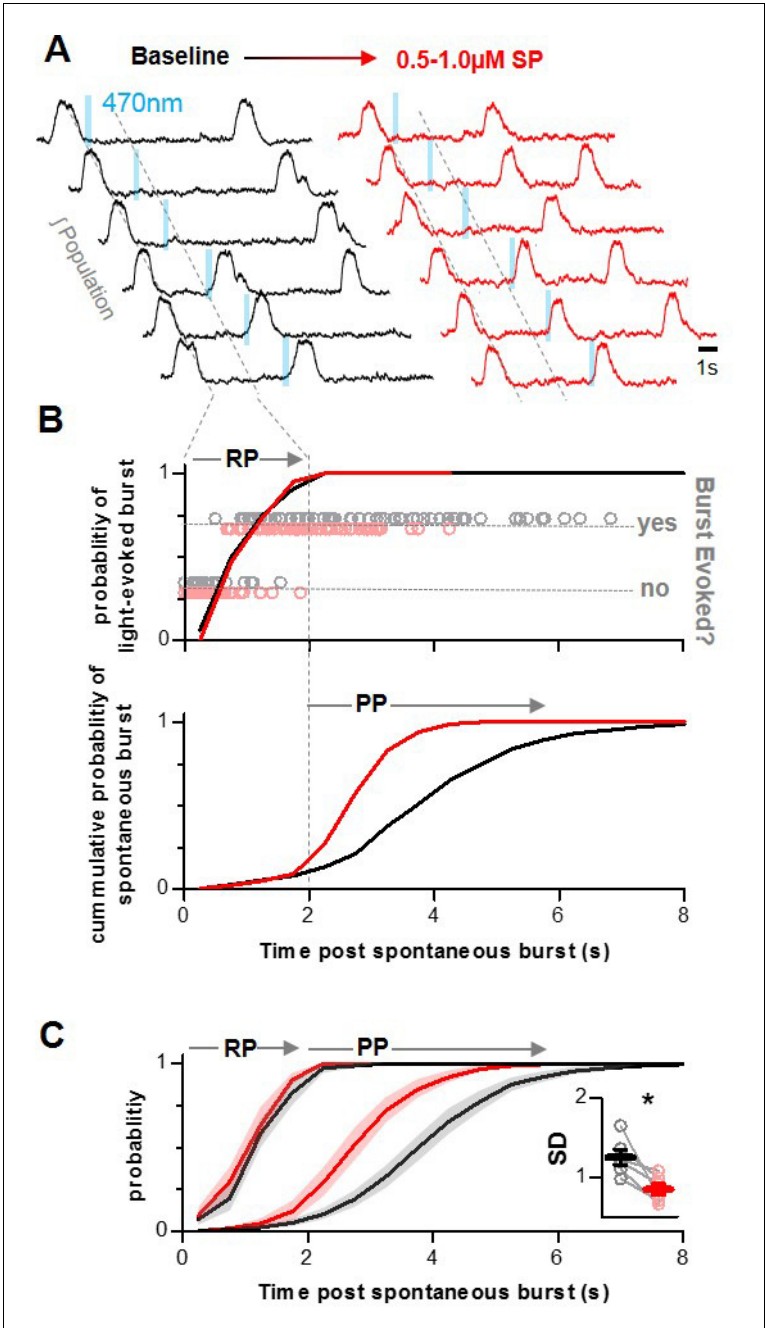

**Figure 1.** Differential modulation of the refractory phase (RP) and recurrent excitation or percolation phase (PP) of the inspiratory rhythm by SP. (**A**) Representative integrated preBötC population recordings from a Dbx1[CreERT2]; Rosa26[ChR2-EYFP] horizontal slice during photostimulation of Dbx1 neurons under baseline conditions (black) and in SP (red). (**B**) Quantified data from the experiment in A showing time-dependent changes in the probabilities of evoking a burst (top) and of a spontaneous burst occurring (bottom). (**C**) Group data from n = 6 experiments. Spontaneous and evoked probability curves were compared under baseline conditions and in SP using non-linear regression analysis. Inset shows the standard deviation (SD) of the inter-burst intervals under baseline conditions and in SP (paired, two tailed t-test). Data available in *Figure 1—source data 1*.

The online version of this article includes the following source data for figure 1:

**Source data 1.** Differential modulation of therefractory phase(RP) and recurrent excitation orpercolation phase(PP) of the inspiratory rhythm by SP.

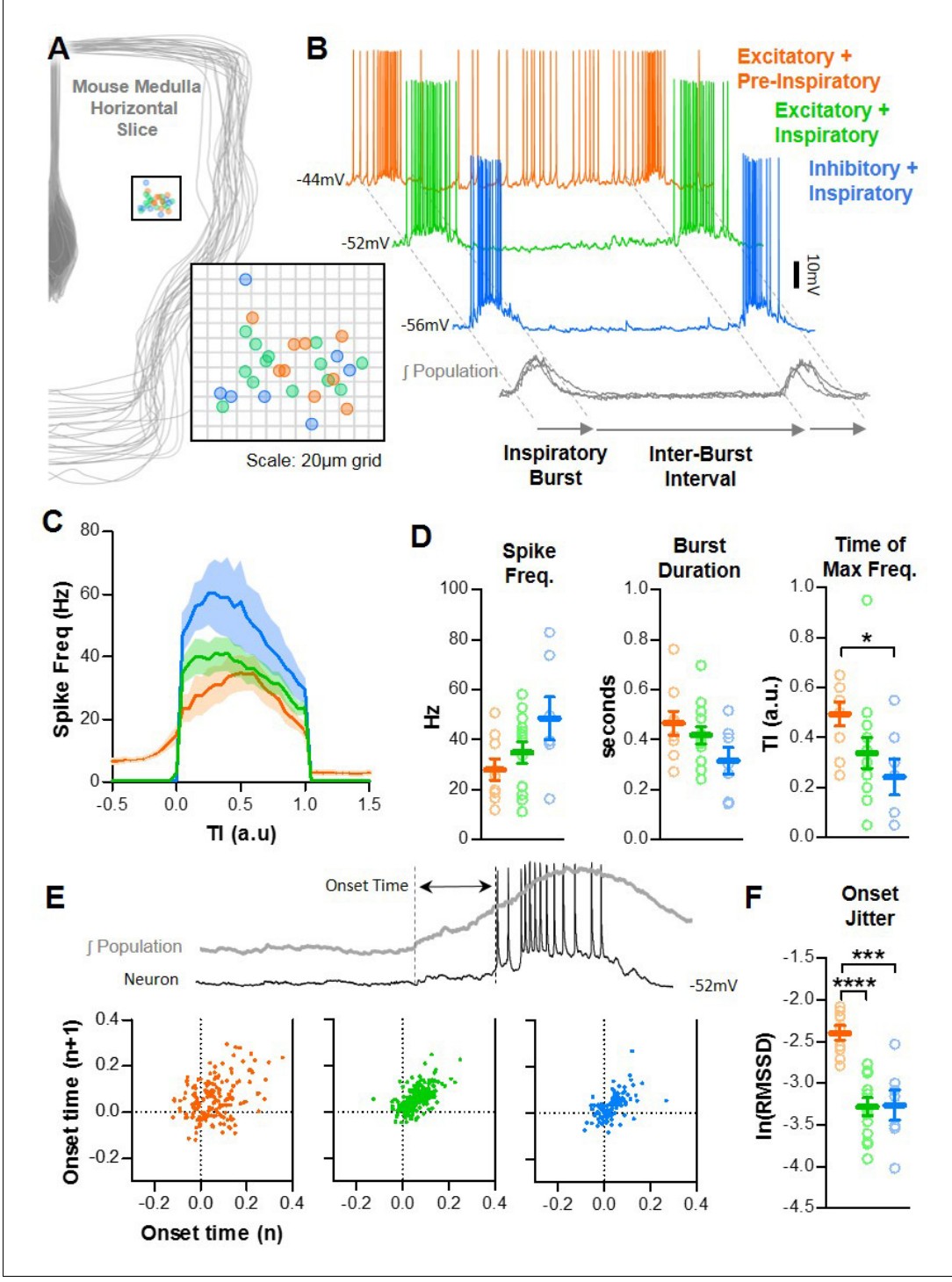

**Figure 2.** Baseline spiking patterns of excitatory and inhibitory neurons in the preBötC. (**A**) Anatomical locations of n = 29 recorded preBötC neurons. (**B**) Example traces of an excitatory neuron with pre-I spiking (orange) and without pre-I spiking (green) and an inhibitory neuron (blue) during the inspiratory burst and inter-burst interval (IBI). (**C**) Quantified spike frequency as a function of time (normalized to burst duration) from n = 9 pre-I excitatory, n = 13 non pre-I excitatory, and n = 7 inhibitory neurons. (**D**) Quantified mean spike frequency, duration, and shape of inspiratory bursts generated by each type of neuron (one-way ANOVA with Bonferroni post hoc tests). (**E**) Example quantification of neuronal burst onset time relative to the preBötC population and Poincaré plots showing onset time variability from n = 9 pre-I excitatory, n = 13 non pre-I excitatory, and n = 7 inhibitory neurons (20 inspiratory bursts/neuron). (**F**) Burst onset time variability or 'jitter' quantified as the natural log of the root

*Figure 2 continued on next page*

*Figure 2 continued*
mean square of successive differences (one-way ANOVA with Bonferroni post hoc tests). Data available in
*Figure 2—source data 1*.
The online version of this article includes the following source data and figure supplement(s) for figure 2:

**Source data 1.** Baseline spiking patterns of excitatory and inhibitory neurons in the preBötC.
**Figure supplement 1.** Representative cell-attached recordings under baseline condition and in SP.
**Figure supplement 2.** Membrane potentials (V$_m$) at baseline and after SP.

inhibitory neurons in the preBötC with pre-I spiking (*Figure 2B*). Excitatory neurons with pre-I spiking (n = 9; 7 WC, 2 CA) had inspiratory spike frequencies ranging from 12.0 to 50.8 Hz (mean: 28.0 ± 4.4 Hz) and burst durations ranging from 272 to 763 ms (mean: 467 ± 48 ms). Similarly, excitatory neurons without pre-I spiking (n = 13; 11 WC, 2 CA) had inspiratory spike frequencies ranging from 11.3 to 58.2 Hz (mean: 34.9 ± 4.3 Hz) and burst durations ranging from 242 to 698 ms (mean: 419 ± 34 ms). Inhibitory neurons (n = 7; 5 WC, 2 CA) also had considerable variability in spike frequency (16.4 to 82.9 Hz; mean: 48.6 ± 8.6 Hz) and burst duration (144 to 517 ms; mean: 316 ± 54 ms) (*Figure 2C, D*).

Inspiratory neuron types also differed in burst shape (*Figure 2C,D*) and burst onset variability (*Figure 2E,F*). Inhibitory neurons had a much more pronounced decrementing spiking pattern than excitatory neurons with maximal spike frequencies occurring at 24 ± 7% of the inspiratory burst duration. Excitatory neurons with pre-I spiking exhibited a rounded burst pattern with maximal spike frequencies occurring at 49 ± 5% of the inspiratory burst duration, whereas excitatory neurons without pre-I spiking were slightly more decrementing with maximal spike frequencies occurring at 34 ± 6% of the inspiratory burst duration. We also quantified the variability in burst onset times – that is the time between the beginning of each preBötC population burst and the onset of the corresponding neuronal burst. In neurons without pre-I spiking, burst onset was defined as the time of the first action potential, whereas in pre-I neurons burst onset was defined as the time with the largest change in action potential frequency. Poincaré plots of onset times for each inspiratory neuron type are shown in *Figure 2E* and burst onset variability (quantified as the natural log of the root mean square of successive differences, ln(RMSSD)) is shown in *Figure 2F*. Overall, average onset times did not differ among neuron types (p>0.05); however, excitatory neurons with pre-I spiking had more cycle-to-cycle variability in burst onset times than excitatory neurons without pre-I spiking or inhibitory neurons (p<0.001).

## Effects of SP on excitatory pre-inspiratory neurons in the preBötC are phase-dependent

Spiking activity of pre-I neurons is expected to contribute to both the refractory and percolation phases of the inspiratory rhythm. During inspiratory bursts, spiking of these neurons contributes to synchronization, which promotes the subsequent refractoriness of the network (*Baertsch et al., 2018*). Following the refractory phase, it is thought that pre-I spiking of these neurons during the IBI facilitates positive-feedback recurrent excitation in the network, which builds up until another inspiratory burst in generated and the cycle restarts (*Del Negro et al., 2018*). Thus, changes in spiking during inspiratory bursts are predicted to alter the duration of the subsequent IBI through modulation of the refractory phase, whereas changes in pre-inspiratory spiking are predicted to alter the IBI by changing the rate of feed-forward excitation during the percolation phase. We examined SP-induced changes in the spiking activity of pre-I neurons during inspiratory bursts and during the IBI. A representative recording is shown in *Figure 3A*. Unexpectedly, SP had little effect on spiking during inspiratory bursts (*Figure 3B*). Changes in burst spike frequency (28.0 ± 4.4 Hz to 29.6 ± 3.8 Hz, p>0.05) and burst duration (467 ± 78 ms to 488 ± 45 ms; p>0.05) were small and inconsistent (*Figure 3B,C*). In contrast, during the IBI, SP increased the average spiking frequency of pre-I neurons from 5.1 ± 1.0 Hz to 9.9 ± 1.8 Hz (p<0.01), and in all cases increased the slope of the pre-inspiratory ramp (average of 1.7 ± 0.8 to 3.7 ± 1.9 Hz/sec), although this did not reach statistical significance (p=0.113) due to the large variability among neurons (*Figure 3D,E*). Changes in IBI spike frequency and slope induced by SP were coincident with a significant decrease in the duration of the IBI (*Figure 3E*). Thus, these phase-dependent changes in spiking activity at the level of individual

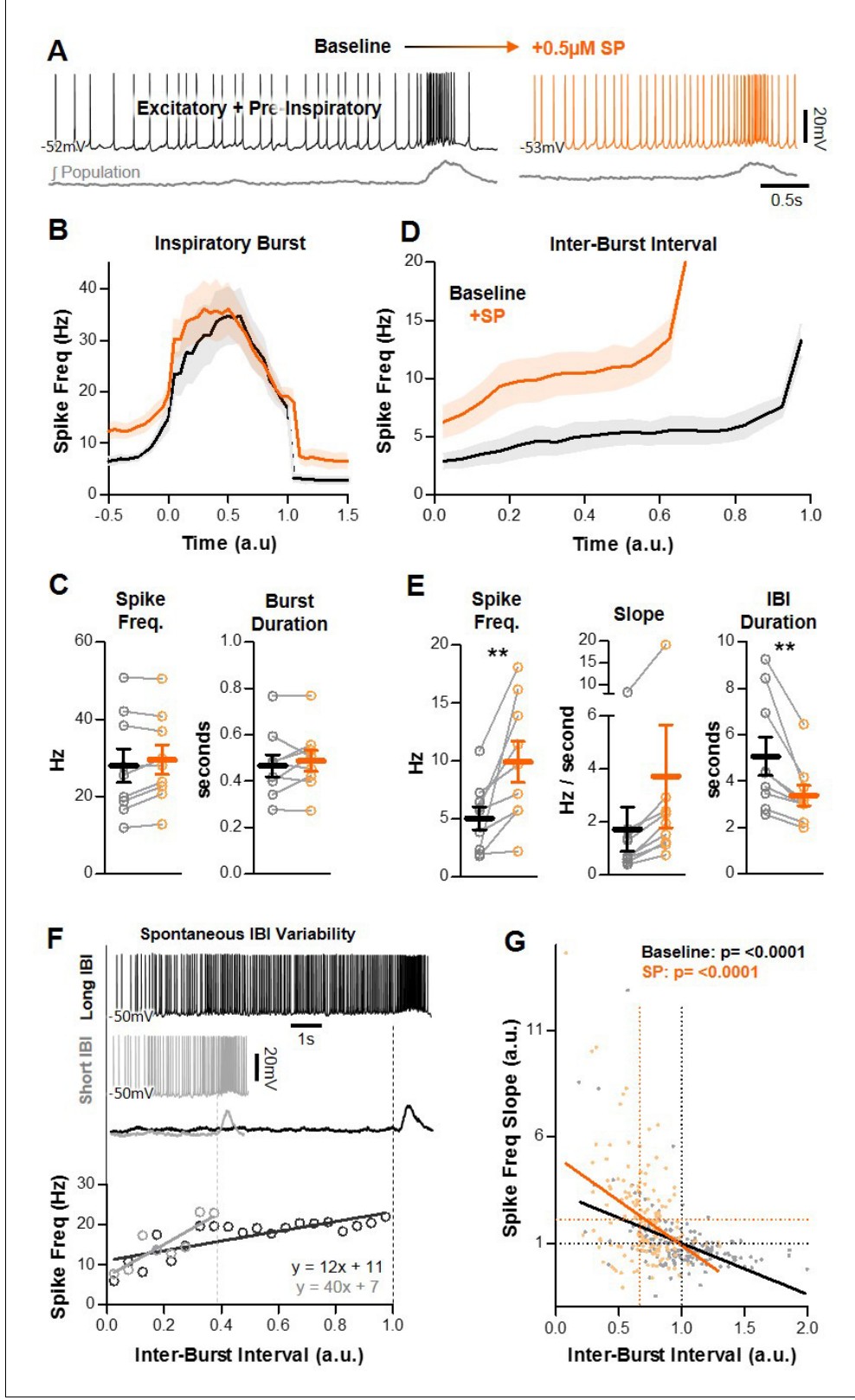

**Figure 3.** Effects of SP on pre-I excitatory neurons in the preBötC. (**A**) Example intracellular recording from a pre-I neuron at baseline (black) and in SP (orange) with corresponding integrated preBötC population activity (gray). (**B**) Quantified spike frequency as a function of time (normalized to inspiratory burst duration) in n = 9 pre-I neurons. (**C**) Mean spike frequency and burst duration of pre-I neurons (paired, two tailed t-tests). (**D**) Quantified spike

*Figure 3 continued*

frequency vs. time (normalized to IBI duration) during the inter-burst interval showing changes in pre-inspiratory ramp activity induced by SP. (E) Mean spike frequency, pre-inspiratory ramp slope, and IBI duration (paired, two tailed t-tests). (F) Example spiking of a pre-I neuron during a long (black) and short (gray) inter-burst interval under baseline conditions (top), and quantified pre-inspiratory slope during each IBI (below). (G) Inverse relationship between the slope of pre-inspiratory spiking and the length of the IBI from n = 9 pre-I neurons (20 consecutive IBIs/neuron) at baseline and in SP (parameters normalized to baseline values) (linear regression analysis). Data available in *Figure 3—source data 1*.

The online version of this article includes the following source data for figure 3:

**Source data 1.** Effects of SP on pre-I excitatory neurons in the preBötC.

pre-I excitatory neurons likely contribute to the differential effects of SP on the refractory and percolation phases observed at the network level (see *Figure 1*).

Since SP also reduced the variability of the IBI at the network level, we examined the relationship between the duration of individual IBIs and the slope of pre-inspiratory spiking activity. An example recording of an excitatory pre-inspiratory neuron during a long IBI (black) and a short IBI (gray), and the quantified spike frequency over the duration of each IBI, is shown in *Figure 3F*. Group data for n = 9 neurons is shown in *Figure 3G*. Twenty consecutive inspiratory cycles were analyzed for each neuron and the duration of each IBI was compared to the pre-inspiratory slope during that cycle. To highlight effects related to cycle-to-cycle variability, values were normalized to the average baseline IBI duration and pre-inspiratory slope for each neuron, respectively. We found that, under control conditions and in SP, there was a significant inverse relationship between the duration of a given IBI and the slope of the corresponding pre-inspiratory ramp, such that pre-inspiratory spiking activity at the level of individual neurons can predict the duration between inspiratory bursts at the network level.

## SP recruits a subpopulation of excitatory preBötC neurons to exhibit pre-inspiratory spiking

Unlike pre-I neurons, excitatory neurons that are silent during the IBI are unable to participate in the feed-forward process of recurrent excitation because they lack pre-inspiratory spiking activity. However, these neurons are expected to contribute to network synchronization during inspiratory bursts, and as a result they have the potential to modulate the refractory period. In response to SP, excitatory neurons that did not spike during the IBI under baseline conditions exhibited two distinct phenotypes. Some (8/13) remained silent during the IBI (teal), whereas others (5/13) developed pre-inspiratory spiking (green) (*Figure 4A*). Excitatory neurons that were not recruited to spike during the IBI (n = 7 WC, 1 CA) also had no change in spike frequency (38.6 ± 5.0 to 37.1 ± 4.9 Hz; p>0.05) or burst duration (430 ± 53 to 428 ± 51 ms, p>0.05) during inspiratory bursts (*Figure 4B,C*), despite a shortened IBI duration (p<0.05). Since these neurons had no change in spiking throughout the inspiratory cycle, it is unlikely that they contribute to SP-induced frequency facilitation of the inspiratory rhythm. In neurons that were recruited to spike during the IBI (n = 4 WC, 1 CA), spiking frequency increased from 0 to 6.6 ± 1.4 Hz (p<0.01). In SP, these neurons exhibited a pre-inspiratory ramp (4.1 ± 1.7 Hz/sec), which was coincident with a shorter IBI duration (p<0.05) (*Figure 4D,E*). During individual inspiratory cycles, the slope of the SP-induced pre-inspiratory ramp had a significant inverse relationship with the duration of the IBI (*Figure 4F*). During inspiratory bursts, spiking patterns did not change in spike frequency (28.9 ± 7.8 to 31.9 ± 6.6 Hz, p>0.05), burst duration (399 ± 28 to 441 ± 36 ms, p>0.05), or burst shape (*Figure 4B,C*). Thus, a subpopulation of non pre-I excitatory neurons develops pre-I activity in SP without a change in spiking activity during bursts, suggesting that the number of neurons that can participate in the percolation phase increases in the presence of SP, without significant effects on the amount of excitation during bursts and the resulting refractory period.

## SP does not change spiking activity of inhibitory inspiratory neurons in the preBötC

In contrast to excitatory neurons, the activity of inhibitory neurons during inspiratory bursts reduces network synchronization and the refractory period (*Baertsch et al., 2018*). Since the RP of the

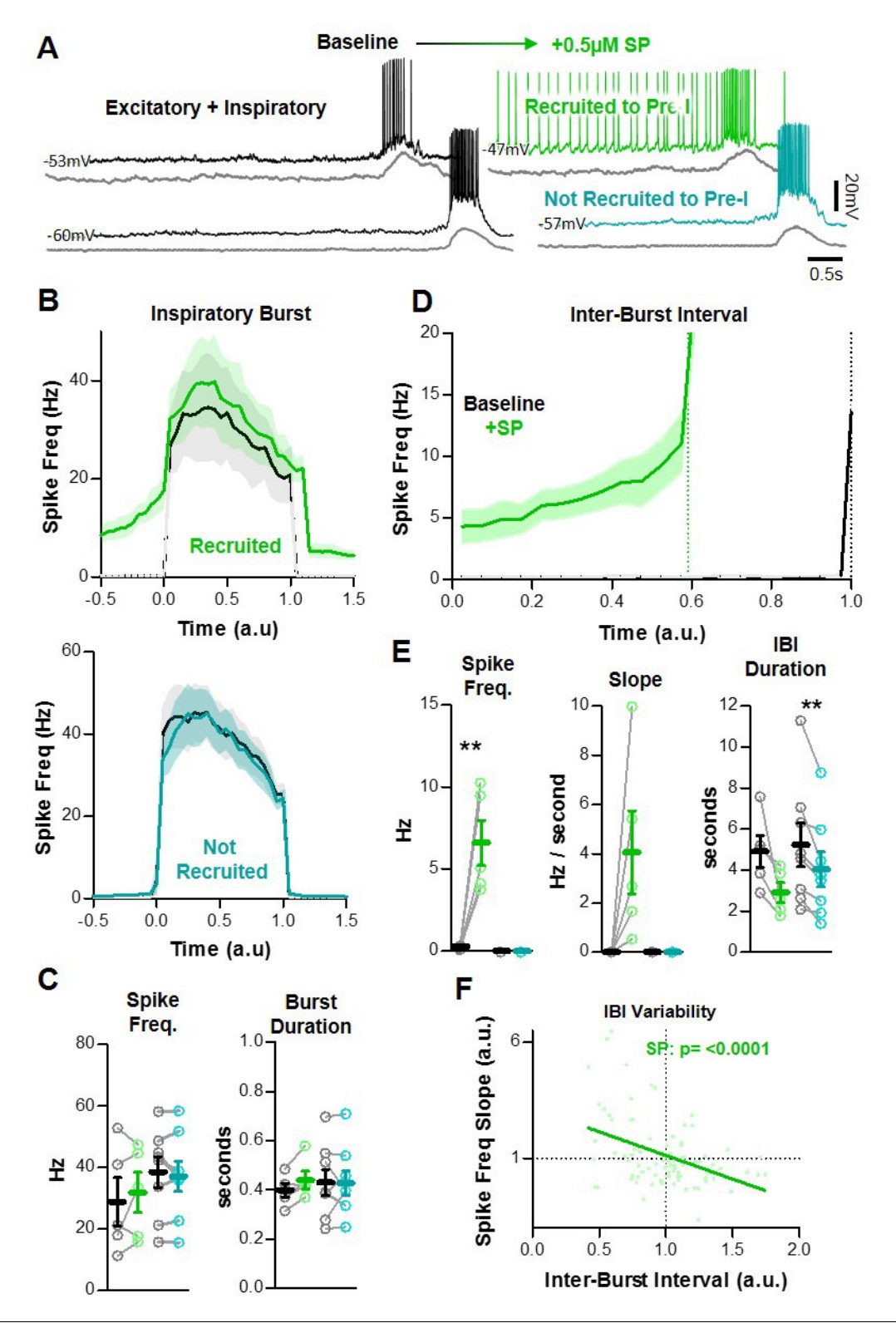

**Figure 4.** SP recruits a subpopulation of excitatory preBötC neurons to participate in the percolation phase. (A) Example intracellular recordings from two excitatory neurons, one that develops pre-I activity in SP (green) and one that does not (teal). Corresponding integrated preBötC population activity is shown below each trace (gray). (B) Quantified spike frequency as a function of time (normalized to baseline inspiratory burst duration) in n = 5

*Figure 4 continued on next page*

*Figure 4 continued*

excitatory neurons that were recruited to pre-I (top) and n = 8 excitatory neurons that were not recruited to pre-I (bottom). (**C**) Mean spike frequency and burst duration in both neuron groups (paired, two-tailed t-tests). (**D**) Quantified spike frequency vs. time (normalized to baseline IBI duration) during the inter-burst interval showing the recruitment of pre-inspiratory ramp activity by SP. (**E**) Mean spike frequency, pre-inspiratory ramp slope, and IBI duration in both neuron groups (paired, two tailed t-tests). (**F**) Inverse relationship between the slope of pre-inspiratory spiking and the length of the IBI in SP from n = 5 excitatory neurons that were recruited to pre-I (20 consecutive IBIs/neuron) (linear regression analysis). Data available in *Figure 4—source data 1*.

The online version of this article includes the following source data for figure 4:

**Source data 1.** SP recruits a subpopulation of excitatory preBötC neurons to participate in thepercolation phase.

---

preBötC network was not changed by SP (see *Figure 1*), we hypothesized that inhibition during bursts would also be unchanged by SP. To test this, we recorded spiking activity from n = 7 (5 WC, 2 CA) inhibitory preBötC neurons under baseline conditions and following application of SP. A representative recording is shown in *Figure 5A*. In the presence of SP, spike frequency and burst duration of inspiratory inhibitory neurons did not change during bursts (48.6 ± 8.6 to 43.5 ± 6.7 Hz, p>0.05; and 316 ± 54 to 328 ± 56 ms, p>0.05, respectively), despite a coincident shortening of the IBI (p<0.05) (*Figure 5B,C*). Spiking activity of inhibitory neurons also did not change during the IBI, since all recorded neurons remained silent between inspiratory bursts. Thus, in response to SP, inhibitory neurons in the preBötC had no change in spiking throughout the inspiratory cycle and are therefore unlikely to play a role in SP-induced facilitation of inspiratory frequency.

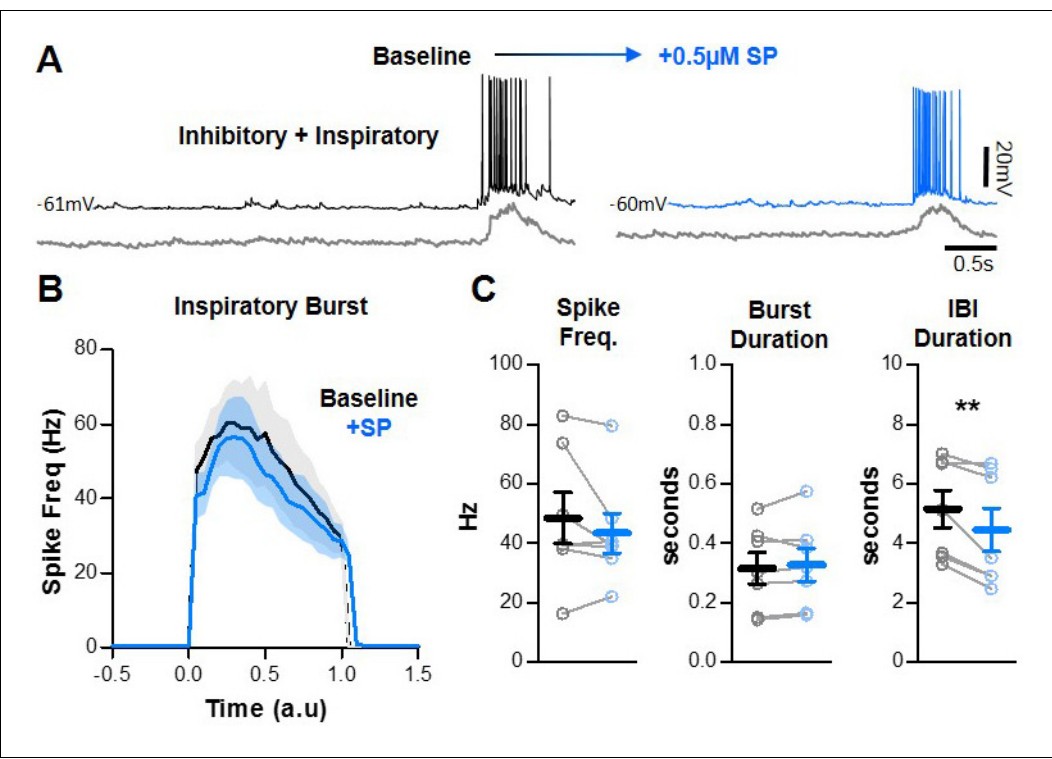

**Figure 5.** SP does not change inhibitory network interactions. (**A**) Example intracellular recordings from an inhibitory preBötC neuron under baseline conditions (black) and in SP (blue) with corresponding integrated preBötC population activity is shown below (gray). (**B**) Quantified spike frequency as a function of time (normalized to baseline inspiratory burst duration) in n = 7 inhibitory neurons. (**C**) Mean spike frequency, burst duration, and IBI (paired, two-tailed t-tests). Data available in *Figure 5—source data 1*.

The online version of this article includes the following source data for figure 5:

**Source data 1.** SP does not change inhibitory network interactions.

## SP increases stochasticity among excitatory preBötC neurons during inspiratory bursts

Next, we sought to unravel potential mechanisms that may prevent SP from causing hyper-synchronization of the preBötC network and increased refractory times. Since synchronization is often reduced with increased stochasticity (*Carroll and Ramirez, 2013*; *Harris et al., 2017*; *Zerlaut and Destexhe, 2017*), we compared the cycle-to-cycle variability in burst onset times (see *Figure 2E*) under baseline conditions and in the presence of SP (*Figure 6*). Although average burst onset times were not significantly altered by SP for any neuronal type ($p>0.05$), SP did have effects on burst onset stochasticity. This is demonstrated as greater dispersions in the Poincaré plots shown in *Figure 6A*. However, these SP-induced changes in burst onset variability (i.e. 'onset jitter') differed across neuronal types. Burst onset jitter of excitatory pre-I neurons, which was generally high under baseline conditions (see *Figure 2F*), did not change significantly in SP ($p>0.05$) (*Figure 6B*). In

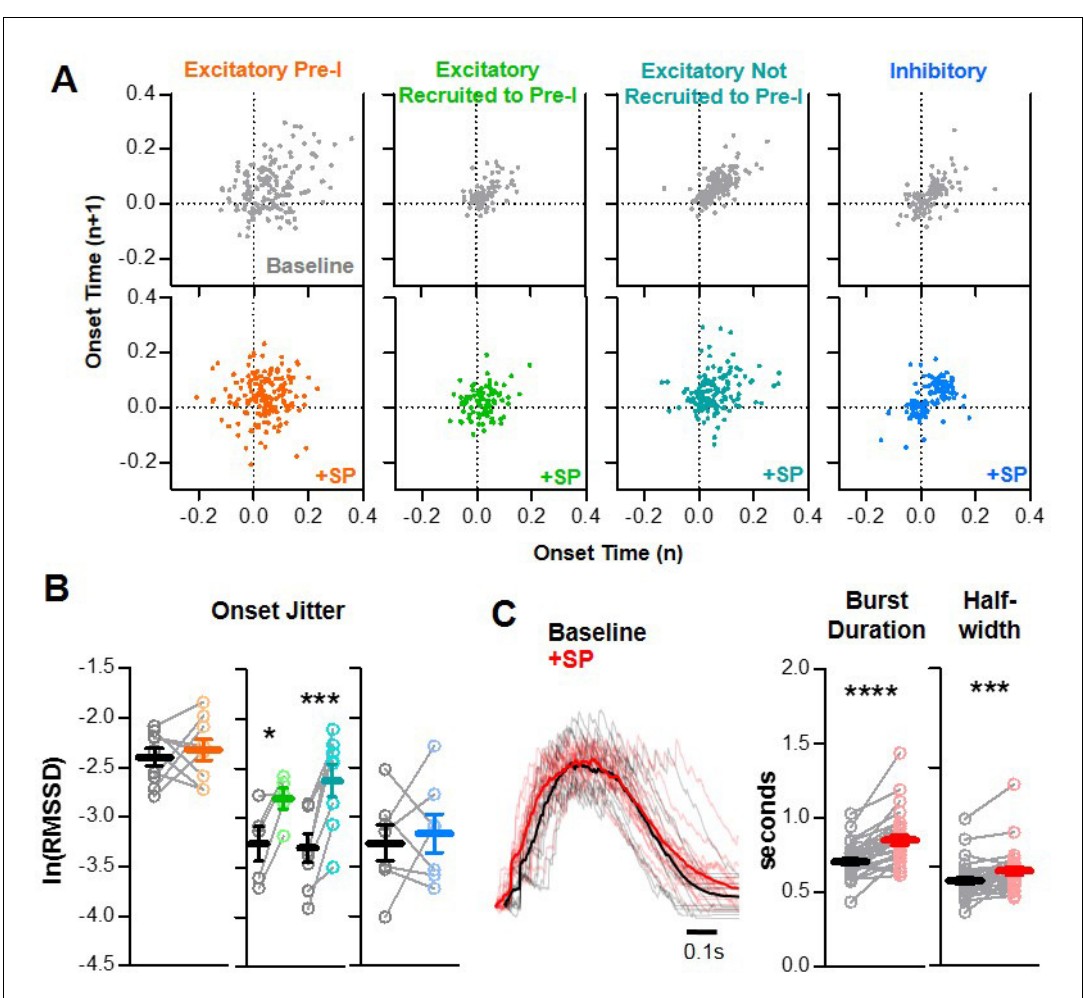

**Figure 6.** SP increases onset variability among non pre-I excitatory neurons during preBötC bursts. (A) Poincaré plots of burst-to-burst variability in onset times under baseline conditions (gray) and in SP for n = 9 excitatory, pre-I (orange); n = 5 excitatory, recruited to pre-I (green); n = 8 excitatory, not recruited to pre-I (teal); and n = 7 inhibitory (blue) neurons. (B) Burst onset time variability or 'jitter', quantified as the natural log of the root mean square of successive differences, at baseline and in SP for each neuron type. (paired, two tailed t-tests). (C) Representative population bursts at baseline (black) and in SP (red) with averaged traces in bold and quantified burst duration and burst half-width from the n = 29 slices in which onset jitter was quantified (paired, two tailed t-tests). Data available in *Figure 6—source data 1*.

The online version of this article includes the following source data for figure 6:

**Source data 1.** P increases onset variability among non pre-I excitatory neurons during preBötC bursts.

contrast, excitatory neurons that did not exhibit pre-I spiking and had relatively low onset jitter under baseline conditions exhibited increased onset jitter in the presence of SP (p<0.05). Among this group of excitatory neurons, burst onset jitter was increased by SP regardless of whether or not the neuron was recruited to develop pre-inspiratory spiking. Inhibitory preBötC neurons, on the other hand, had relatively inconsistent changes in burst onset variability induced by SP, with no change in mean onset jitter (p>0.05). These results suggest that SP increases the stochasticity of burst onset in a subgroup of preBötC excitatory neurons. Indeed, this increase in stochasticity at the level of individual neurons was reflected in the activity of the preBötC network as a slight increase in the duration (707 ± 22 to 854 ± 33 ms; p<0.0001) and halfwidth (429 ± 18 to 478 ± 20 ms; p<0.0001) of population bursts in SP (*Figure 6C*).

## Inspiratory neurons rostral to the preBötC have heterogeneous responses to SP

Substance P increase the frequency and stability of the inspiratory rhythm in transverse slices that isolate the preBötC from the rest of the ventral respiratory column (VRC) (*Gray et al., 1999*; *Peña and Ramirez, 2004b*). However, neurons with inspiratory activity are not confined to the preBötC but are distributed along the VRC (*Barnes et al., 2007*; *Zuperku et al., 2019*). Indeed, the inspiratory network seems to be spatially dynamic since excitatory neurons located rostral to the preBötC can be conditionally recruited to participate in the inspiratory rhythm (*Baertsch et al., 2019*). This rostral expansion of the active inspiratory network is associated with an increase in the excitation/inhibition ratio, longer refractory times, and slower inspiratory frequencies (*Baertsch et al., 2019*). Therefore, we explored whether the opposite may also occur: Could the size of the active inspiratory network shrink, and could this be another mechanism that prevents SP from causing increased excitation during inspiratory bursts? To test this, we recoded spiking activity from n = 16 (13 WC, 3 CA) inspiratory neurons located in the rostral VRC (n = 5 excitatory, n = 7 inhibitory, n = 4 unknown). The anatomical locations of these neurons relative to the preBötC neurons described above are shown in *Figure 7A*. Overall, spiking activity patterns and responses to SP were less consistent among rostral neurons than preBötC neurons. To convey this heterogeneity, spike rasters for each rostral neuron over 20 consecutive inspiratory bursts are shown in *Figure 7B,C,D*. Despite this variability, spiking frequency during bursts was reduced by SP in all (5/5) rostral excitatory neurons (−55.6 ± 13.5% change from 11.1 ± 4.7 to 6.7 ± 4.0 Hz; p<0.05) (*Figure 8A,C*), whereas changes were inconsistent among inhibitory rostral neurons with no change on average (39.5 ± 34.7% change from 5.3 ± 1.5 to 6.1 ± 1.5 Hz; p>0.05) (*Figure 8B,C*). Thus, the potential contribution of rostral excitatory neurons to synchronization of the inspiratory rhythm was reduced by SP, while inhibitory influences were relatively unchanged. Burst onset variability of both excitatory and inhibitory inspiratory neurons rostral of the preBötC was generally high under baseline conditions, and it was further increased by SP (p<0.05) (*Figure 8D,E*).

During the inter-burst interval, the spiking activity of rostral neurons was considerably different from neurons in the preBötC. Unlike the pre-I spiking described for excitatory neurons in the preBötC (see *Figures 3* and *4*), excitatory rostral neurons were generally silent during the IBI under baseline conditions, and they remained silent following application of SP (*Figure 7B*). In contrast, four out of seven inhibitory rostral neurons exhibited spiking during the IBI under baseline conditions, and in three of these neurons spike frequency during the IBI was increased by SP. Among the three inhibitory rostral neurons that were silent during the IBI under baseline conditions, two were recruited to spike during the IBI in response to SP. Overall, six of seven exhibited spiking during the IBI in SP (1.5 ± 0.6 Hz at baseline to 3.5 ± 1.2 Hz in SP; p>0.05) (*Figure 8F*). The spiking of these neurons did not exhibit a pre-inspiratory ramp under baseline conditions (slope: 0.11 ± 0.14 Hz/second; p>0.05) or in the presence of SP (slope: 0.08 ± 0.15 Hz/second; p>0.05), suggesting changes in the spiking of these inhibitory neurons during the IBI is not be regulated by excitatory pre-I neurons in the preBötC.

## Discussion

The rhythm generating network that produces breathing movements must constantly adjust to changing metabolic demands and also adapt to overlapping volitional and reflexive behaviors (*Feldman et al., 2013*; *Ramirez and Baertsch, 2018b*). Unravelling mechanisms that support this

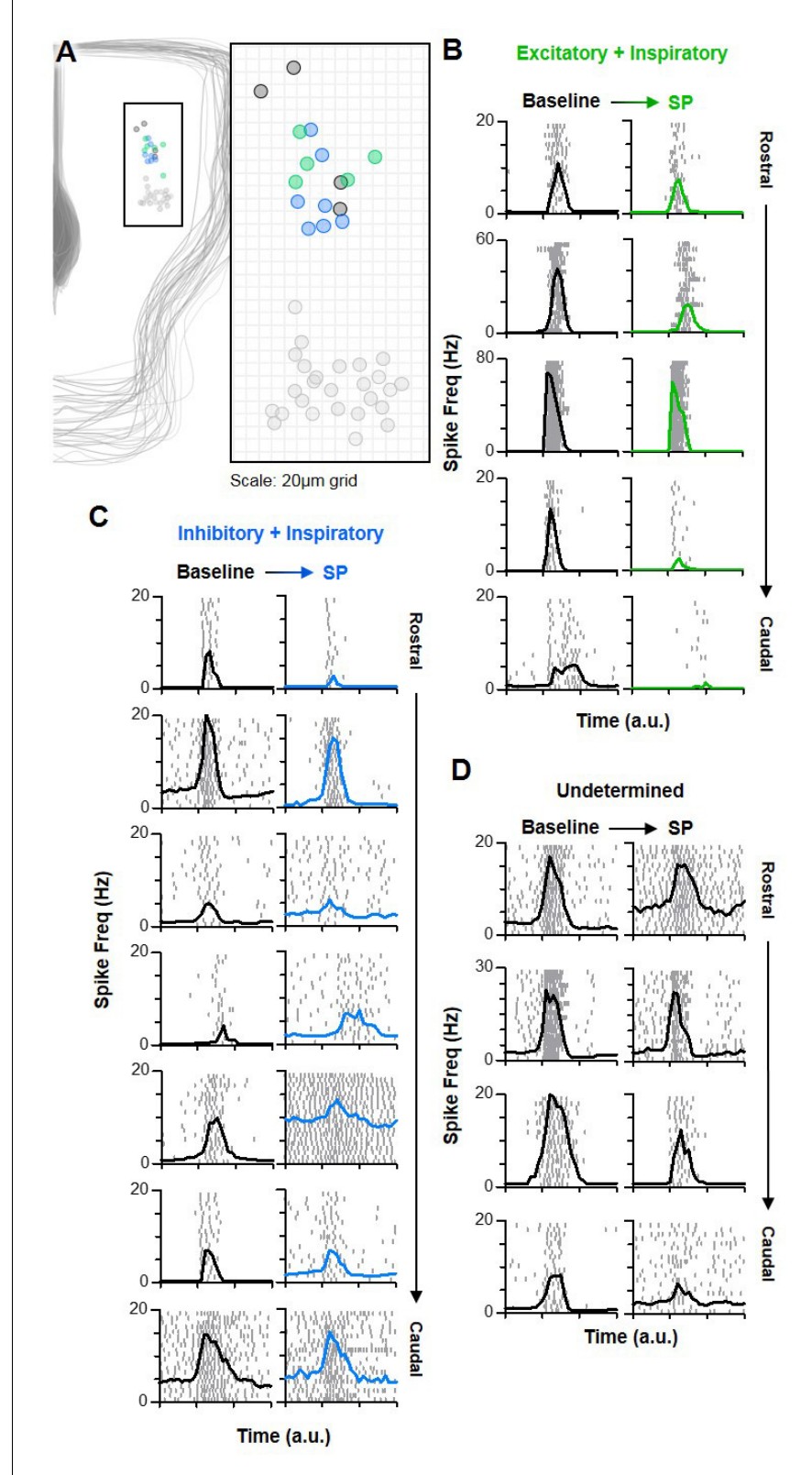

**Figure 7.** Inspiratory neurons rostral of the preBötC have varied responses to SP. (A) Anatomical locations of n = 16 rostral inspiratory neurons [n = 5 excitatory (green), n = 7 inhibitory (blue), and n = 4 unknown (black)], relative to preBötC neurons (light gray). (B–D) Spike rasters (each row is one burst cycle; 20 consecutive cycles are

*Figure 7 continued on next page*

stacked) and average spike frequency for each excitatory (B), inhibitory (C), and unknown (D) rostral neuron at baseline and in SP. Time is normalized to the preBötC population burst duration, denoted by the x-axis tick marks.

dynamic control may improve our understanding of disorders of the nervous system that destabilize

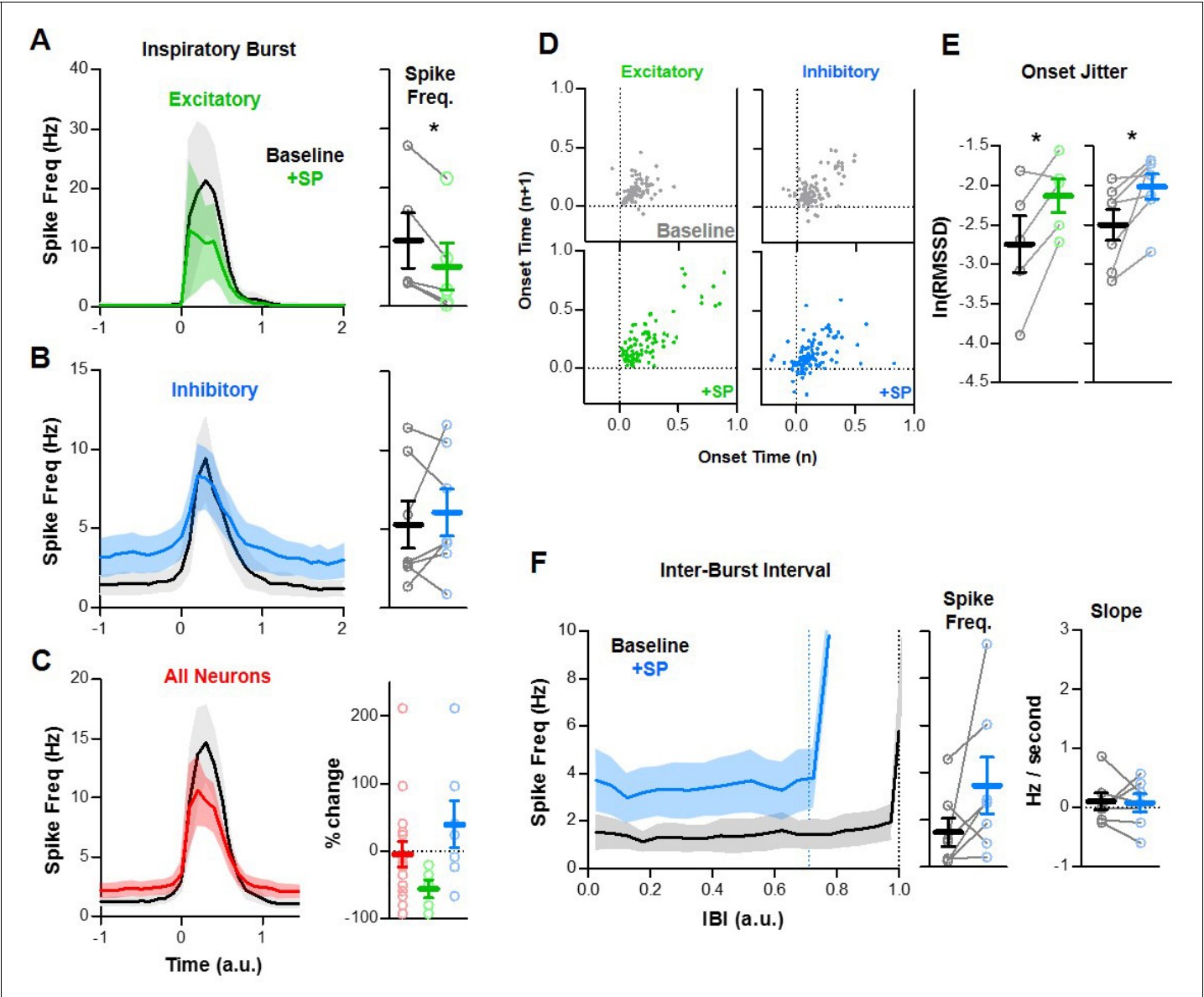

**Figure 8.** Quantified effects of SP on rostral excitatory and inhibitory inspiratory neurons. (A–C) Average spike frequency as a function of time (normalized to preBötC population burst duration) in n = 5 excitatory neurons (A), n = 7 inhibitory neurons (B), and in all rostral neurons (n = 16, red). Mean changes in inspiratory spike frequency at baseline and in SP are plotted to the right of each panel (paired, two tailed t-tests, A and B; one-way ANOVA, (C). (D) Poincaré plots showing burst-to-burst variability in onset times among rostral excitatory (n = 5) and inhibitory (n = 7) neurons (20 consecutive inspiratory bursts/neuron). (E) Mean burst onset time variability or 'jitter' at baseline and in SP for each excitatory (left) and inhibitory (right) neurons (paired t-tests). (F) Average spike frequency of rostral inhibitory neurons during the inter-burst interval (normalized to baseline IBI) at baseline and in SP. Mean changes in spike frequency and slope during the IBI are plotted to the right (paired, two tailed t-tests). Data available in *Figure 8— source data 1*.

The online version of this article includes the following source data for figure 8:

**Source data 1.** Quantified effects of SP on rostral excitatory and inhibitory inspiratory neurons.

breathing, such as Parkinson's disease, Rett syndrome, sudden infant death syndrome, congenital central hypoventilation syndrome, multiple-systems atrophy, and amyotrophic lateral sclerosis (*Oliveira et al., 2019*; *Ramirez et al., 2018*; *Schwarzacher et al., 2011*; *Katz et al., 2009*; *Moreira et al., 2016*). Changes in neuromodulatory systems within the brainstem have been linked to many of these and other respiratory control disorders (*Doi and Ramirez, 2008*; *Viemari et al., 2005*). Here, we introduce the concept that distinct phases of the rhythmogenic process can be differentially regulated by neuromodulation to dynamically control the frequency and stability of breathing (*Figure 9*).

During inspiration, each burst is assembled stochastically via heterogeneous interactions among a combination of intertwined synaptic and intrinsic properties (*Ramirez and Baertsch, 2018b*). Although exclusively excitatory synaptic interactions (*Kam et al., 2013a*) or intrinsic bursting mechanisms (*Peña et al., 2004a*) may be able to produce rhythm in isolation, this is unlikely to occur under normal conditions since these properties interact strongly. To the contrary, the combination of excitatory and inhibitory synaptic interactions with intrinsic bursting properties, known as the 'rhythmogenic triangle' (*Ramirez and Baertsch, 2018b*), is critical for the flexibility of this dynamic network (*Ramirez et al., 2004*; *Rubin and Smith, 2019*; *Smith et al., 2000*; *Richter and Smith, 2014*). Neuromodulators play important roles in regulating both synaptic and intrinsic bursting properties, perhaps best demonstrated in invertebrate model systems. In these networks, neuromodulators can inhibit or strengthen synaptic interactions (*Harris-Warrick et al., 1998*; *Marder et al., 2014*;

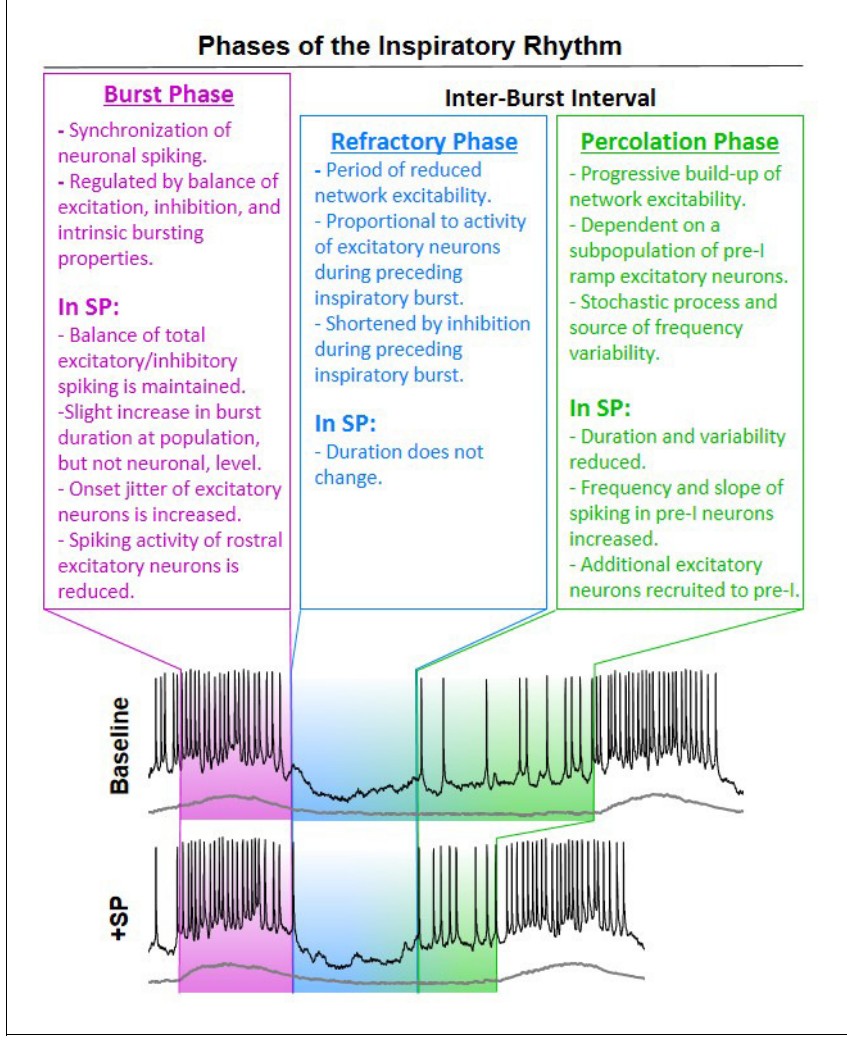

**Figure 9.** Intrinsic phases of the inspiratory rhythm generator and their modulation by substance P.

*Nusbaum et al., 2001*), as well as induce or suppress intrinsic bursting properties (*Elson and Selverston, 1992*; *Flamm and Harris-Warrick, 1986*; *Zhang and Harris-Warrick, 1994*). These important principles also apply to the aminergic and peptidergic modulation of the mammalian preBötC network (*Doi and Ramirez, 2010*). For example, in the absence of synaptic interactions, the activity of both $I_{NaP}$ and $I_{CAN}$-dependent autonomous bursting neurons in the preBötC is modulated by SP (*Ben-Mabrouk and Tryba, 2010*; *Peña and Ramirez, 2004b*). Although it is known that $NK_1R$ expressing preBötC neurons are important for rhythmogenesis (*Gray et al., 2001*; *Guyenet and Wang, 2001*), it is unclear how these modulatory effects regulate the frequency and stability of the inspiratory rhythm. Evidence suggests that $NK_1R$ is primarily expressed on excitatory neurons (*Gray et al., 1999*), including those with pre-I activity (*Guyenet and Wang, 2001*). However, if SP activates excitatory neurons during inspiratory bursts and facilitates their intrinsic bursting properties, one would expect a prolongation of the refractory phase, which would limit rather than promote a frequency increase. Instead, we found that SP does not affect the spiking frequency of either excitatory or inhibitory preBötC neurons during the inspiratory burst phase (*Figures 3*, *4* and *5*). Because local synaptic interactions among interconnected excitatory and inhibitory neurons is a fundamental characteristic of the preBötC network (*Ramirez and Baertsch, 2018b*), we infer that the changes in neuronal spiking, or lack thereof, that we observed under current clamp conditions reflect local synaptic interactions within the preBötC. Accordingly, we suggest that the balance of excitatory and inhibitory inputs to preBötC neurons is not changed by SP, although voltage-clamp experiments under visual control would be warranted to more directly assess this balance. Nonetheless, an important question is: How is the activity of inspiratory neurons, and presumably the balance of excitation and inhibition, maintained during bursts despite the excitatory effects of SP?

One possibility is that excitatory inputs originating from outside the preBötC could be reduced to prevent SP from causing hyperexcitation during inspiratory bursts. Inspiratory neurons located rostral to the preBötC contribute to the dynamic regulation of breathing frequency. Indeed, transverse brainstem slice preparations that lack rostral inspiratory neurons generate less robust rhythms with shorter refractory periods and faster frequencies than horizontal slices (*Baertsch et al., 2019*). On the other hand, recruitment of these neurons is associated with increased excitation during inspiratory bursts, a prolonged refractory phase, and consequently slower breathing frequencies (*Baertsch et al., 2019*). However, under normal conditions, inhibition restrains the rhythm generating ability of these rostral neurons allowing faster, more dynamic breathing. Interestingly, we found that many inhibitory neurons rostral of the preBötC exhibit tonic spiking activity during the IBI, which was moderately increased by SP. In contrast, rostral excitatory neurons did not spike during the IBI, and spiking was only slightly suppressed during inspiratory bursts (*Figure 8*). Thus, unlike their role in modulating other dynamic breathing responses such as gasping (*Baertsch et al., 2019*), the contribution of rostral inspiratory neurons to SP-induced frequency increase seems to be relatively modest. Indeed, $NK_1R$ is preferentially expressed in the preBötC compared to other regions of the ventral respiratory column (*Gray et al., 1999*), which may explain the relatively heterogeneous effects of SP on the activity of rostral neurons (*Figure 7*).

Another possibility is that synchronization of excitatory neurons is reduced during inspiration, thereby preventing hyperexcitation during the burst phase in response to SP. The assembly of inspiratory bursts is known to be stochastic, which is observed as significant variability in the cycle-to-cycle onset time of inspiratory neuron bursts relative to the preBötC population (*Carroll and Ramirez, 2013*). This variability in activation order, or onset jitter, is related, at least in part, to the sparse functional connectivity within the preBötC network (*Carroll and Ramirez, 2013*). We found that SP increases the variability of burst onset times among inspiratory neurons located within and rostral to the preBötC. This increase in timing variability was most pronounced among excitatory neurons, specifically those that lack pre-I activity (*Figure 6*). Thus, although other mechanisms such as depletion of excitatory synaptic vesicles (*Rubin et al., 2009*) may also contribute, we propose that increased jitter among excitatory neurons impairs synchronization of the network and helps maintain the balance of excitation and inhibition during inspiratory bursts in the presence of SP. Future computational modeling studies will be important to directly test this hypothesis.

The inspiratory burst phase is followed by a period of reduced excitability in the preBötC network. This refractory period is thought to arise from a combination of presynaptic depression (*Kottick and Del Negro, 2015*) and activation of slow hyperpolarizing current(s) (*Baertsch et al., 2018*; *Krey et al., 2010*) in glutamatergic Dbx1 neurons during inspiratory bursts. Indeed,

refractoriness is maximal immediately following the inspiratory burst followed by a gradual recovery of excitability (*Figure 1*), likely as vesicles are recycled and hyperpolarizing conductances are inactivated. This refractory phase manifests experimentally as a period during which the probability of evoking an ectopic inspiratory burst via optogenetic stimulation of Dbx1 neurons is reduced (*Figure 1A*). However, the refractory period is not absolute as it can be overcome if the stimulus is of sufficient strength (*Vann et al., 2018*). Using a stimulus procedure consistent with previous reports (*Baertsch et al., 2018*; *Baertsch et al., 2019*; *Kottick and Del Negro, 2015*), we found that SP does not change the duration of the refractory period. This finding is consistent with our demonstration that excitatory neurons do not show an increased activation during the inspiratory burst phase (*Figures 3B* and *4B*). Importantly, the minimum duration of spontaneous inter-burst intervals continues to be restrained by the refractory period in SP (*Figure 1*). Thus, refractory mechanisms remain an important determinant of breathing frequency in the presence of this neuromodulator. This may be functionally important to prevent excitatory neuromodulators such as SP from driving the respiratory network out of its physiological frequency range. Indeed, neuronal networks must not only be capable of dynamically regulating their frequency, but they must also be able to maintain stability in spite of heterogeneous, intrinsically variable cellular components that receive converging inputs from numerous excitatory neuromodulators (*Marder et al., 2014*).

As the inspiratory network transitions out of the refractory phase, some excitatory neurons, presumably with more depolarized basal membrane potentials (*Butera et al., 1999*; *Smith et al., 2000*), begin to fire action potentials prior to the onset of the next synchronized population burst (*Figure 3*). The associated synaptic interactions, which may particularly involve Dbx1 neurons (*Wang et al., 2014*), are thought to stochastically percolate through a network of recurrently connected excitatory neurons (*Del Negro et al., 2018*). This network-based process, sometimes referred to as the 'group pacemaker' hypothesis (*Del Negro and Hayes, 2008*), likely occurs simultaneously with changes in intrinsic membrane properties such as $I_{NaP}$ (*Butera et al., 1999*; *Ramirez et al., 2016*; *Smith et al., 2000*), that together lead to a gradual build-up of excitability within the network. Consequently, the spiking activity of excitatory pre-I neurons between inspiratory bursts exhibits a ramping pattern. We found that the magnitude and slope of this pre-inspiratory ramp, as well as the number of excitatory neurons that have pre-inspiratory activity, is increased by SP. These recruited excitatory neurons also exhibit a pronounced pre-inspiratory ramp, suggesting that they participate in the recurrent excitation process. Together, these effects likely increase the rate of recurrent excitation during this percolation phase to accelerate the onset of the next burst (*Figure 9*), underling the increase in breathing frequency induced by SP. Moreover, we found that, on a cycle-to-cycle basis, the slope of pre-inspiratory ramp activity is inversely related to the inter-burst interval. Thus, variations in the stochastic percolation of excitation during this phase seem to predict variability in the duration between inspiratory bursts. Our data suggest that, by increasing the rate of recurrent excitation, this process becomes more consistent and breathing irregularity is reduced. We conclude that the dual effects of SP on breathing frequency and stability are primarily a consequence of its effects on the percolation phase of the inspiratory rhythm.

Based on our collective results, we propose a conceptual framework for inspiratory rhythm generation in which three distinct phases, the burst phase, refractory phase, and percolation phase, can be differentially modulated to influence breathing dynamics and stability (*Figure 9*). Although these three phases seem to be intrinsic to the inspiratory rhythm generator, we anticipate that non-inspiratory neurons and/or respiratory related brain regions outside the preBötC (e.g. *Anderson et al., 2016*; *Fu et al., 2019*; *Huckstepp et al., 2018*; *Richter and Smith, 2014*) can provide important regulatory control over each phase of the inspiratory cycle, leading to predictable phase-dependent effects on rhythm dynamics. This concept may provide a foundation for understanding breathing in the context of many other physiological and pathological conditions (*Bright et al., 2018*; *Saito et al., 2001*), and it may also serve as a guide for understanding the dynamic control rhythm generating networks in general.

# Materials and methods

## Animals

All experiments and animal procedures were approved by the Seattle Children's Research Institute's Animal Care and Use Committee (approved protocol #15981) and conducted in accordance with the National Institutes of Health guidelines. Experiments were performed on neonatal (p6-p12) male and female C57-Bl6 mice bred at Seattle Children's Research Institute (SCRI). All mice were group housed with access to food and water ad libitum in a temperature controlled (22 ± 1℃) facility with a 12 hr light/dark cycle. For optogenetic experiments, Vglut2$^{Cre}$ (*Slc17a6*) and Vgat$^{Cre}$ (*Slc32a1*) (*Vong et al., 2011*) homozygous breeder lines were obtained from Jackson Laboratories (Stock numbers 028863 and 016962, respectively). Heterozygous Dbx1$^{CreERT2}$ mice were donated by Dr. Del Negro (College of William and Mary, VA) and a homozygous breeder line was generated at SCRI. Dbx1$^{CreERT2}$ dams were plug checked and injected at E9.5 with tamoxifen (24 mg/kg, i.p.) to target preBötC neurons (*Kottick et al., 2017*). Cre mice were crossed with homozygous mice containing a floxed STOP channelrhodopsin2 fused to an EYFP (Ai32) reporter sequence (JAX #024109). Male and female offspring were chosen at random based on litter distributions.

## In vitro medullary horizontal slice preparations

Horizontal medullary slices were prepared from postnatal day 6–12 mice as described in detail previously (*Anderson et al., 2016*; *Baertsch et al., 2019*). Brainstems were dissected in ice cold artificial cerebrospinal fluid (aCSF; in mM: 118 NaCl, 3.0 KCl, 25 NaHCO$_3$, 1 NaH$_2$PO$_4$, 1.0 MgCl$_2$, 1.5 CaCl$_2$, 30 D-glucose) equilibrated with carbogen (95% O$_2$, 5% CO$_2$). When equilibrated with gas mixtures containing 5% CO$_2$ at ambient pressure, aCSF had an osmolarity of 300–312mOSM and a pH of 7.40–7.45. The dorsal surface of each brainstem was secured with super glue to an agar block cut at a ~ 15° angle (rostral end facing up). Brainstems were first sectioned in the transverse plane (200 μm steps) using a vibratome (Leica 1000S) until the VII nerves were visualized. The agar block was then reoriented to position the ventral surface of the brainstem facing up with the rostral end toward the vibratome blade to section the brainstem in the horizontal plane. The blade was leveled with the ventral edge of the brainstem and a single ~850 μm step was taken to create the horizontal slice.

Slices were placed in a custom recording chamber containing circulating aCSF (~15 ml/min) warmed to 30℃. The [K+] in the aCSF was then gradually raised from 3 mM to 8 mM over ~10 min to boost neuronal excitability. Rhythmic extracellular neuronal population activity was recorded by positioning polished glass pipettes (<1 MΩ tip resistance) filled with aCSF on the surface of the slice. Signals were amplified 10,000X, filtered (low pass, 300 Hz; high pass, 5 kHz), rectified, integrated, and digitized (Digidata 1550A, Axon Instruments). The activity of single neurons was recorded using the blind patch clamp approach. Recording electrodes were pulled from borosilicate glass (4–8 MΩ tip resistance) using a P-97 Flaming/Brown micropipette puller (Sutter Instrument Co., Novato, CA) and filled with intracellular patch electrode solution containing (in mM): 140 potassium gluconate, 1 CaCl$_2$, 10 EGTA, 2 MgCl$_2$, 4 Na$_2$ATP, and 10 Hepes (pH 7.2). To map the location of recorded neurons, patch pipettes were backfilled with intracellular patch solution containing 2 mg/ml Alexa Fluor568 Hyrdrazide (ThermoFisher). Neuronal spiking activity was recorded in whole-cell or cell-attached configuration with a multiclamp 700B amplifier in current clamp mode (Molecular Devices, Sunnyvale, CA). Although recordings were performed in 'current-clamp' mode, artificial currents were not applied to control spiking activity or to shift membrane potential from its normal resting value. However, in some cases, a small holding current (0 to −20 pA) was applied for leak compensation. Although, in all experiments, recording settings used to obtain baseline values were maintained constant throughout the experiment, in some recordings electrode potential drift precluded analysis of membrane potentials. Extracellular and intracellular signals were acquired in pCLAMP software (Molecular Devices, Sunnyvale, CA). Immediately following electrophysiology experiments, fresh, unfixed slices were imaged to determine the location(s) of the intracellular recording sites.

## Optogenetic and pharmacological manipulations

We chose the *Dbx1*$^{CreERT2}$;*Rosa26*$^{ChR2-EYFP}$ mouse line to assess the refractory period at the network level because it more specifically targets the neurons that are presumed to be rhythmogeneic in the preBötC (*Vann et al., 2018*). However, we have previously shown that the refractory period in

preBötC slices determined using $Dbx1^{CreERT2}$;$Rosa26^{ChR2-EYFP}$ is not different than the refractory period determined using $Vglut2^{Cre}$;$Rosa26^{ChR2-EYFP}$ (*Baertsch et al., 2018*). A 200 µm diameter glass fiber optic (0.24NA) connected to a blue (470 nm) high-powered LED was positioned above the pre-BötC contralateral to the extracellular electrode and ipsilateral to the intracellular electrode. Power was set ≤1 mW/mm$^2$. To determine the probability of light-evoking inspiratory bursts, 200 ms light pulses were TTL-triggered every 20 s to stimulate Dbx1 neurons (≥50 trials per experiment). Trials were excluded from the analysis if the light pulse occurred during an ongoing spontaneous inspiratory burst. During most intracellular recordings, neurons were classified as excitatory or inhibitory using an optogenetic approach. Because not all excitatory neurons in the preBötC are derived from Dbx1 progenitors, we used a combination of the $Vglut2^{Cre}$;$Rosa26^{ChR2-EYFP}$ and $Vgat^{Cre}$;$Rosa26^{ChR2-EYFP}$ mouse lines to identify individual neurons as excitatory or inhibitory. In $Vgat^{Cre}$;$Rosa26^{ChR2-EYFP}$ slices, neurons that depolarized during photostimulation were classified as inhibitory, while those that hyperpolarized or did not respond were presumed to be excitatory. Because a depolarizing response to stimulation of excitatory neurons could be driven synaptically instead of from channelrhodopsin2 expression directly, in $Vglut2^{Cre}$;$Rosa26^{ChR2-EYFP}$ and $Dbx1^{CreERT2}$;$Rosa26^{ChR2-EYFP}$ slices, neurons were classified as excitatory or inhibitory based on the presence or absence of a depolarizing response to light, respectively, following pharmacological blockade of excitatory AMPAR- and NMDAR-dependent synaptic transmission (20 µM CNQX, 20 µM CPP).

Substance P was purchased from Tocris (Cat#: 1156), diluted in water to a concentration of 5 mM, and stored in stock aliquots at −20°C. In all experiments, a ~ 10 min baseline period of stable inspiratory activity was recorded prior to bath application of substance P to 0.5–1.0 µM. Intracellular and extracellular population activity was then recorded for >10 min prior to washout into fresh aCSF.

## Microscopy

2.5X brightfield and epifluorescent images of the dorsal surface of horizontal slices were acquired on a Leica DM 4000 B epifluorescence microscope. Following intracellular recording experiments, the location of each recorded neuron within the horizontal slice was immediately quantified by overlaying the brightfield and an epifluorescent image of Alexa Fluor 568 labeled cell(s). Images were then traced in powerpoint and overlaid with the midline and rostral edge (VII nerve) aligned to show the relative locations of recorded cells (see *Figures 2A* and *7A*).

## Statistical analysis

Data was analyzed using Clampfit software (Molecular Devices). Integrated population bursts and individual action potentials were detected using Clampfit's peak-detection analysis. Statistical analyses were performed using GraphPad Prism6 software and are detailed for each experiment in the Figure Legends. Groups were compared using appropriate two-tailed t-tests, or one-way ANOVAs with Bonferonni's multiple comparisons post hoc tests. Welch's correction was used for unequal variances where appropriate. Non-linear regression analysis was used to determine differences between probability curves (*Figure 1*). Linear regression analyses were used to determine relationships between inter-burst intervals and pre-inspiratory ramp slope (*Figures 3G* and *5F*). Differences were considered significant at $p < 0.05$ and data are displayed as individual data points with overlaid means ± SE. Significance is denoted in the figures as follows: ****$p < 0.0001$; ***$p < 0.001$; **$p < 0.01$; *$p < 0.05$. Experimenters were not blinded during data collection or analysis.

## Acknowledgements

We thank NIH grants K99 HL145004 (Awarded to NAB), F32 HL134207 (Awarded to NAB), R01 HL126523 (Awarded to JMR), R01 HL144801 (Awarded to JMR), and P01 HL 090554 (Awarded to JMR) for funding this project.

## Additional information

### Competing interests
Jan-Marino Ramirez: Reviewing editor, *eLife*. The other author declares that no competing interests exist.

### Funding

| Funder | Grant reference number | Author |
|---|---|---|
| National Heart, Lung, and Blood Institute | R01 HL126523 | Jan-Marino Ramirez |
| National Heart, Lung, and Blood Institute | R01 HL144801 | Jan-Marino Ramirez |
| National Heart, Lung, and Blood Institute | K99 HL145004 | Nathan A Baertsch |
| National Heart, Lung, and Blood Institute | F32 HL134207 | Nathan A Baertsch |
| National Heart, Lung, and Blood Institute | P01 HL090554 | Jan-Marino Ramirez |

The funders had no role in study design, data collection and interpretation, or the decision to submit the work for publication.

### Author contributions
Nathan A Baertsch, Conceptualization, Formal analysis, Supervision, Funding acquisition, Validation, Investigation, Visualization, Methodology; Jan-Marino Ramirez, Supervision, Funding acquisition

### Author ORCIDs
Nathan A Baertsch https://orcid.org/0000-0003-1589-5575
Jan-Marino Ramirez https://orcid.org/0000-0002-5626-3999

### Ethics
Animal experimentation: All experiments and animal procedures were approved by the Seattle Children's Research Institute's Animal Care and Use Committee and conducted in accordance with the National Institutes of Health guidelines (approved protocol #15981).

### Decision letter and Author response
Decision letter https://doi.org/10.7554/eLife.51350.sa1
Author response https://doi.org/10.7554/eLife.51350.sa2

## Additional files

### Supplementary files
• Transparent reporting form

### Data availability
All data generated during this study are included in the manuscript and supporting files. Source data files have been provided for all figures.

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
