## [Decision Letter]

**Acceptance summary:**

This is a technically impressive study that effectively combines optogenetic, electrophysiological, and pharmacological techniques to analyze cell type specific neuromodulatory actions of substance P (SP) on excitatory and inhibitory neurons in the inspiratory rhythm generator within the mouse preBötzinger complex in vitro. Important novel results are presented suggesting that the demonstrated effects of SP in augmenting inspiratory frequency and rhythm stability are a consequence of its selective effects on the pre-inspiratory recurrent excitation or "percolation" phase of the inspiratory rhythmic cycle where SP is shown to augment the spiking activity and recruit of excitatory pre-I spiking neurons. The important conceptual advance is that neuromodulation of inspiratory cycle dynamics at least in vitro may be understood by considering the respiratory cycle to consist of three distinct phases (inspiratory burst, refractory, and percolation phases) that can be differentially regulated by physiologically important neuromodulators, with predictable phase-dependent effects on rhythm dynamics. The results presented from this study with SP effectively support and emphasize this concept.

**Decision letter after peer review:**

Thank you for submitting your article "Insights into the dynamic control of breathing revealed through cell-type-specific responses to substance P" for consideration by *eLife*. Your article has been reviewed by three peer reviewers, and the evaluation has been overseen by Ronald Calabrese as the Senior and Reviewing Editor. The following individuals involved in review of your submission have agreed to reveal their identity: Muriel Thoby-Brisson (Reviewer #1); Jeffrey C Smith (Reviewer #2).

The reviewers have discussed the reviews with one another and the Reviewing Editor has drafted this decision to help you prepare a revised submission.

Essential revisions:

All the major concerns of the expert reviewers should be addressed in revision. To help guide this process we offer a few consensus comments.

1) There was general concern that the authors cannot really make definitive statements about how SP influences excitatory/inhibitory balance without voltage clamp experiments that directly assess excitatory and inhibitory synaptic input. In the absence of such experiments, it would be satisfactory if the authors present this concept as an inference/speculation with adequate discussion in the manuscript, including further clear discussion about why they think analyses of modulation of more rostral excitatory and inhibitory neuron activity is critical in terms of understanding circuit interactions relevant to the problem, and also indicate the need to perform more comprehensive analyses of synaptic interactions based on direct measurements of synaptic currents/potentials to resolve this issue. Some measurements in the transverse slice would also help clarify if the more complex distributed circuit interactions that the authors seem to be invoking are important. According to the authors' model, the preBötC should be "over-excited" with substance P and measurements of this should include a change burst rate and amplitude. These changes should be then compared to the same dose of SP in the horizontal preparation. The model predicts that SP will have different magnitude of effects in these two networks.

2) There is general concern that the current clamp data needs clear supporting data. (i) Reviewer #2's requirements to provide much more detail about the recordings is essential. (ii) The authors must demonstrate that the data in whole cell recording mode is reproduced by a reasonable number of cell-attached recordings. (iii) The authors also must be clear about how they are defining burst onset. Is it based on the first action potential?

3) The was concern about the issue of "jitter" of burst onset and how the authors measure jitter and about their interpretation of it as a potential mechanism to prevent hypersynchronization. The authors show data that indicating that SP doesn't change the burst duration or spiking frequency for individual cells, but instead increases the "jitter" of burst onset. The authors claim that this is a mechanism to prevent hypersynchronization (subsection “SP increases stochasticity among excitatory preBӧtC neurons during inspiratory bursts”), which is only one potential model and interpretation. If this is included, it must be clearly stated as a model and be limited to the Discussion in the paper. Furthermore, if the burst onset is more variable, then we should expect to see overall changes in burst duration, yet no data on burst characteristics is documented. This should be described and reported.

Reviewer #1:

The very elegant work proposed by Baertsch and Ramirez aims to decipher the cellular mechanisms by which Substance P (SP, a well-known modulator of the respiratory network activity) dynamically controls respiratory rhythmogenesis through cell-type specific effects. They found that SP modulates breathing frequency and stability through an action during a very specific phase of the respiratory cycle: the percolation phase (recurrent excitation phase), but not by changing the inspiratory phase nor the refractory period. In addition, they provide evidences for a cell-specific action as this modulatory mechanism involves preferentially excitatory respiratory neurons capable of exhibiting pre-inspiratory spiking under SP influence without any significant contribution of inhibitory or pre-inspiratory neurons. Thus, they conclude that phase-specific and cell-type specific action of SP is a key mechanism to dynamically control the dynamic of the respiratory network.

Overall the paper is well written, easy to follow and data are nicely illustrated and convincing. I just have a few comments:

1) As stated in the Discussion it is considered that rhythmogenic mechanisms within the respiratory network, supported by the group-pacemaker hypothesis, involve interactions between synaptic and specific membrane properties (such as pacemaker properties) among respiratory neurons. Recording respiratory pacemaker neurons remain challenging due to their poor representativeness. But I was wondering whether the authors had the opportunity to perform their experiments in such a type of respiratory neurons. This could be particularly interesting as, to my knowledge, pacemaker neurons rarely exhibit pre-inspiratory discharge. I would be curious to know whether SP would change this pattern in pacemaker neurons and consequently whether modulation of this specific cell-type could also account for the overall modulatory effect of SP.

2) Do the authors have any indications on the cellular mechanisms underlying the change in discharge pattern of the excitatory neurons in the presence of SP? Is it an increase in recurrent synaptic inputs, changes in membrane properties associated to specific conductances, or both? In other words, do the authors possess data in voltage-clamp mode to reveal possible modulation of either of these features?

Reviewer #2:

This is a technically impressive study that effectively combines optogenetic, electrophysiological, and pharmacological techniques to analyze cell type specific neuromodulatory actions of substance P (SP) on excitatory and inhibitory neurons in the inspiratory rhythm generator within the mouse preBötzinger complex in vitro. Important novel results are presented suggesting that the demonstrated effects of SP in augmenting inspiratory frequency and rhythm stability are a consequence of its selective effects on the pre-inspiratory recurrent excitation or "percolation" phase of the inspiratory rhythmic cycle where SP is shown to augment the spiking activity and recruit of excitatory pre-I spiking neurons. The important conceptual advance is that neuromodulation of inspiratory cycle dynamics at least in vitro may be understood by considering the respiratory cycle to consist of three distinct phases (inspiratory burst, refractory, and percolation phases) that can be differentially regulated by physiologically important neuromodulators, with predictable phase-dependent effects on rhythm dynamics. The results presented from this study with SP effectively support and emphasize this concept.

1) The authors suggest that the inspiratory phase is regulated by the balance of inhibition, excitation, and intrinsic bursting properties of active neurons, and that in the present experiments with SP, the balance of excitation-inhibition is maintained. While the former is certainly the case in general, excitatory and inhibitory inputs to any neuron in the present experiments have not been directly measured (e.g., by voltage clamp measurements) to know for sure about this balance. Similarly, the authors state, but have not directly shown, that SP shifts the balance of excitation and inhibition among rostral inspiratory neurons, which implies that they have performed some analyses of synaptic interactions among neurons in the network, including in preBötC circuits. This needs to be qualified further in the text. I agree that it is important that the data presented show that SP does not change spiking frequency of excitatory inspiratory neurons during the burst phase as well as activity of inhibitory neurons (and by inference probably local inhibitory interactions) during this phase. Perhaps the modulation of leak conductances by SP (noted in the Introduction) can cause depolarizing shifts in neuronal membrane potentials to promote pre-I spiking without altering overall spiking profiles during the inspiratory burst phase.

2) The baseline membrane potentials of neurons before (and therefore) after SP are not indicated in any of the figures (Figure 2, 3, 4, 5) with whole-cell recordings and provided in the text. This information is important and should be provided to understand why the subpopulation of pre-I excitatory neurons is active during the inter-burst interval (presumably they have more depolarized baseline potentials?), the amount of depolarization if any with SP, and why specific neuron types are recruited or not to spike during the IBI interval. It has been postulated for many years that neurons with more depolarized basal membrane potentials exhibit pre-I spiking activity that is important for rhythmogenesis by providing synaptic excitation during the pre-I spiking phase (e.g., Butera et al., 1999).

Reviewer #3:

The authors confirm that SP decreases the interval between respiratory bursts and show this shortening is due to shrinking of a later portion of the interval that they dub the "percolation" phase. This point is cleanly and clearly demonstrated in Figure 1, but this contribution alone does not substantially deepen our understanding of respiratory rhythm generation and modulation. Remaining experiments are purely descriptive, flawed with technical issues, and do not provide the proposed mechanistic understanding. Given Yeh et al., 2017, defined the molecular mechanism for SP modulation of breathing, this work does not provide any advancement necessitating publication to a general audience as is.

1) One primary focus of the paper is to record from previously defined populations of pre-inspiratory and inspiratory excitatory neurons and inspiratory inhibitory neurons to determine if SP has cell-type specific effects. However, a major technical flaw throughout the manuscript is the use of current clamp recordings instead of cell attached recordings. Although the authors claim to have done both in the Materials and methods, no cell-attached recordings are shown. In current clamp, the experimenter has complete control of the cells resting membrane potential and can therefore easily make any cell look pre-inspiratory or inspiratory. With this flaw in mind, the authors cannot then claim to have identified cell-type specific modulation by substance P.

2) In Figure 2E/F and Figure 6 the authors focus attention to the effect of substance P on the onset of neural bursting activity versus preBötC population bursting. They claim that SP increases variability in this timing for pre-inspiratory neurons and that this is key to ensure the preBötC burst size does not increase. As shown in Figure 2E, the onset of the burst appears arbitrarily defined and as shown, would be nearly impossible to definitively decide for a pre-inspiratory neuron. Furthermore, this point is influenced by what membrane potential the experimenter is holding the neuron at in current clamp.

3) Many years of papers have shown that SP increases the preBötC rhythm in a coronal slice preparation. This result shows that brainstem regions rostral to the preBötC are not required for SP modulation of preBötC activity. However, in Figures 7/8 the authors record from neurons rostral to the preBötC and observe changes in activity after SP. They claim these changes cause a shift to increased inhibitory neuron activity that helps to maintain preBötC excitation. This is never directly measured and there are too many logical leaps here, such as:

i) Do these rostral neurons project to the preBötC?

ii) Are there decreased excitatory inputs into the preBötC?

iii) Does bursting in the preBötC become "over-excited" in a coronal slice?

iv) If SP acts through NALCN to depolarize cells, how is it decreasing excitatory neural activity? Given the lack of understanding, these figures add confusion.

[Editors' note: further revisions were requested prior to acceptance, as described below.]

Thank you for submitting your article "Insights into the dynamic control of breathing revealed through cell-type-specific responses to substance P" for consideration by *eLife*. Your article has been reviewed by three peer reviewers, and the evaluation has been overseen by Ronald Calabrese as the Senior and Reviewing Editor. The following individuals involved in review of your submission have agreed to reveal their identity: Muriel Thoby-Brisson (Reviewer #1); Jeffrey C Smith (Reviewer #2).

The reviewers have discussed the reviews with one another and the Reviewing Editor has drafted this decision to help you prepare a revised submission.

The authors have done an admirable job responding to the previous review. One reviewer asks for further clarifications, which should be seriously considered. When the authors re-submit no external further review will be necessary.

Essential revisions:

The revised manuscript has nicely incorporated the feedback from the three essential revisions. The inclusion of the membrane potential in the featured recordings as well as the number of cell-attached recordings performed addresses essential revision comment #2. Although the N is low for the number of cell-attached recordings, this data is important to confirm that a subset of excitatory-inspiratory neurons display pre-I activity only after SP application. Additionally, as the authors point out, the described membrane potentials suggest that the recruited neurons are likely those with higher resting membrane potentials before SP application. With this in mind, the authors should consider providing the membrane potential before and after SP for all of the recorded neurons of the various neural types. While the depolarization and recruitment of cells into pre-I firing is interesting, it will be essential in the future to determine if the recruitment is required for the SP induced shortening of the percolation phase. Alternative models still remain, like the increased activity of pre-I neurons alone drives the shortening of the percolation phase. The additional description of the methods will enhance reproducibility by others and the modified discussion nicely handles how the various observations made in the manuscript may result in the observed selective effect of SP to shorten the percolation phase and addresses essential revision comments #1 and #3.

---

## [Author Response]

Essential revisions:All the major concerns of the expert reviewers should be addressed in revision. To help guide this process we offer a few consensus comments.1) There was general concern that the authors cannot really make definitive statements about how SP influences excitatory/inhibitory balance without voltage clamp experiments that directly assess excitatory and inhibitory synaptic input. In the absence of such experiments, it would be satisfactory if the authors present this concept as an inference/speculation with adequate discussion in the manuscript, including further clear discussion about why they think analyses of modulation of more rostral excitatory and inhibitory neuron activity is critical in terms of understanding circuit interactions relevant to the problem, and also indicate the need to perform more comprehensive analyses of synaptic interactions based on direct measurements of synaptic currents/potentials to resolve this issue. Some measurements in the transverse slice would also help clarify if the more complex distributed circuit interactions that the authors seem to be invoking are important. According to the authors' model, the preBötC should be "over-excited" with substance P and measurements of this should include a change burst rate and amplitude. These changes should be then compared to the same dose of SP in the horizontal preparation. The model predicts that SP will have different magnitude of effects in these two networks.

We have added discussion of the limitation of current clamp and that we are inferring synaptic interactions from spiking activity. Follow up voltage clamp experiments are also suggested in the text. See responses to reviewers 1 and 2 for further discussion.

We have not included additional experiments in transverse slices because (i) based on the variable effects of SP on rostral neurons we observed in the horizontal slice, we do not anticipate that changes in the activity of the rostral inspiratory column are a large contributor to SP-induced frequency facilitation. This is consistent with the known distribution of NK1 receptors, the expression of which is less abundant outside the preBötC. The Discussion has been revised to better emphasize this. And (ii) the horizontal slice and transverse slice have different baseline properties including the duration of the refractory period, burst amplitude, burst duration and burst frequency (Baertsch et al., 2019), which may make interpretation of the suggested experiments challenging. Moreover, adding any meaningful experiment in transverse slices would mean repeating all the reported experiments in this preparation. See response to reviewer 3 for further discussion.

2) There is general concern that the current clamp data needs clear supporting data. (i) Reviewer #2's requirements to provide much more detail about the recordings is essential. (ii) The authors must demonstrate that the data in whole cell recording mode is reproduced by a reasonable number of cell-attached recordings. (iii) The authors also must be clear about how they are defining burst onset. Is it based on the first action potential?

We have addressed reviewer 2’s concerns, and now include V_m_ information for representative whole-cell recordings. We have added the numbers of whole-cell vs. cell-attached recordings in the text and now include example cell-attached recordings in Figure 2—figure supplement 1. We now clearly define how burst onset was determined in both pre-I and non pre-I neurons in the Results section.

3) The was concern about the issue of "jitter" of burst onset and how the authors measure jitter and about their interpretation of it as a potential mechanism to prevent hypersynchronization. The authors show data that indicating that SP doesn't change the burst duration or spiking frequency for individual cells, but instead increases the "jitter" of burst onset. The authors claim that this is a mechanism to prevent hypersynchronization (subsection “SP increases stochasticity among excitatory preBӧtC neurons during inspiratory bursts”), which is only one potential model and interpretation. If this is included, it must be clearly stated as a model and be limited to the Discussion in the paper. Furthermore, if the burst onset is more variable, then we should expect to see overall changes in burst duration, yet no data on burst characteristics is documented. This should be described and reported.

We now present this explicitly as a hypothesis in the Discussion. We also analyzed changes in the duration of population bursts in SP and found that there is a small but significant increase in burst duration as well as burst half-width. This information is now included in the results and Figure 6C.

Reviewer #1:[…] Overall the paper is well written, easy to follow and data are nicely illustrated and convincing. I just have a few comments:1) As stated in the Discussion it is considered that rhythmogenic mechanisms within the respiratory network, supported by the group-pacemaker hypothesis, involve interactions between synaptic and specific membrane properties (such as pacemaker properties) among respiratory neurons. Recording respiratory pacemaker neurons remain challenging due to their poor representativeness. But I was wondering whether the authors had the opportunity to perform their experiments in such a type of respiratory neurons. This could be particularly interesting as, to my knowledge, pacemaker neurons rarely exhibit pre-inspiratory discharge. I would be curious to know whether SP would change this pattern in pacemaker neurons and consequently whether modulation of this specific cell-type could also account for the overall modulatory effect of SP.

Based on our previous observations (Carroll and Ramirez, 2013), we assume that many, but not all, pre-I neurons are in fact bursting pacemaker neurons. In the study by Carroll and Ramirez, we characterized the cycle-to-cycle activation of more than 600 neurons that were recorded extracellularly (i.e. without manipulating their membrane potential). We demonstrate that the probability of discharging before the population burst (i.e. pre-I) in the network, was strongly related (<0.0001) to overall spike rate and interspike interval metric of burstiness in synaptic block (<0.0001). We also found that mean spike rate was significantly correlated with mean spike rate after synaptic block – which meant that bursting neurons that have high spike rates after synaptic block have also higher spike rates in the network and were more likely to have pre-inspiratory activity. Of course, in this extracellular and pre-optogenetic study we did not discriminate between inhibitory and excitatory neurons. In the Carroll and Ramirez study we also demonstrated that not all bursting pacemaker neurons had pre-inspiratory activity, and in fact many bursting neurons also discharged later in the burst. Based on the present study, which showed that inhibitory neurons have no pre-inspiratory activity, it is tempting to speculate that bursting neurons that discharged at or after burst onset included inhibitory neurons, while those with pre-I activity were more likely excitatory neurons, but as stated above, this was not determined in this study. Assuming that bursting pacemaker neurons that discharged before the population bursts were excitatory neurons, one could also speculate that ectopic bursting in pacemakers may contribute to mechanisms of recurrent excitation and the percolation phase. Indeed, previous work from our lab has shown that SP can facilitate bursting in both I_NaP_ and I_CAN_ pacemaker neurons, which we include in the Discussion (second paragraph). But, since in the present study we did not specifically test for pacemaker properties under synaptically isolated conditions, we can only infer and not make specific conclusions about how intrinsic bursting contributes to the pre-I percolation phase. What, our data indicate however, is that despite the known modulation of intrinsic bursting, substance P does not change spiking activity of excitatory neurons during bursts when respiratory neurons are integrated within the network.

2) Do the authors have any indications on the cellular mechanisms underlying the change in discharge pattern of the excitatory neurons in the presence of SP. Is it an increase in recurrent synaptic inputs, changes in membrane properties associated to specific conductances, or both? In other words, do the authors possess data in voltage-clamp mode to reveal possible modulation of either of these features?

This is an important question. Since it has been previously shown that some preBӧtC neurons depolarize in response to SP (e.g. Yeh et al., 2017; Pena and Ramirez, 2004) including pacemakers (Pena and Ramirez, 2002), we anticipate that intrinsic depolarization of some of these neurons can facilitate spiking activity, and thereby promote mechanisms of recurrent excitation But, in addition to I_NaP_ and I_CAN_, it is also likely that the Nalcn current (Yeh et al., 2017) as well as NMDA mechanisms (Pena and Ramirez, 2004) can contribute to this depolarization. While unraveling which of all the contributing currents is most important is an interesting question, we feel this is beyond the scope of the present study. Thus, we did not apply substance P following blockade of synaptic transmission to assess intrinsic membrane responses. Instead, our aim was to understand the consequences of such a depolarization for the activity of the network as a whole. Hence, we focused on the spiking activity of individual neurons that were identified as excitatory or inhibitory. Our rationale was that a depolarization that does not result in a change in spiking activity will have negligible consequences for the function of the network, whereas a neuron that changes its spiking activity *will* have consequences for network activity, irrespective of which underlying current caused the increased spiking activity, and irrespective of whether the neuron is intrinsically modulated by Substance P or indirectly via synaptic inputs.

Moreover, we expect bath application of SP to have a tonic influence on NK_1_R expressing neurons – e.g. causing a persistent shift in V_m_ under steady state conditions. Thus, although intrinsic changes in membrane potential and increased recurrent synaptic activity are likely linked as described above, we anticipate that the time-dependent changes in spiking or pre-I ramp during the IBI is generated by a build-up of recurrent synaptic activity. Indeed, we found that many rostral inhibitory/inspiratory neurons have spiking during the IBI, but they do not exhibit a time-dependent pre-I ramp, suggesting they are not involved in the recurrent excitation mechanism.

We did not perform voltage clamp experiments to directly assess synaptic inputs. This was intentional because we felt that using the VC approach would not significantly contribute to our understanding of cell-type-specific inputs, since we assume that EPSCs in any given neuron will likely originate from many different neurons. Hence, it would be difficult to quantify which inputs would come from pre-inspiratory neurons vs. from inspiratory neurons without a pre-inspiratory component. Also, just as the post-synaptic targets are unknown when monitoring spiking activity, the locations of pre-synaptic neurons are unknown when monitoring EPSCs in voltage clamp. Instead, we assume that the cell-type specific changes in spiking we observed in current clamp will involve changes in synaptic inputs in other preBӧtC neurons since this network is known to contain prevalent recurrent connections. Indeed, this is a requirement for the network to synchronize. To what extent the changes in synaptic input are driven by changes in leak currents, potassium currents, persistent sodium currents and TRPM, TRPC currents is an important question, but again, we do not expect to obtain a simple answer as all of those currents will somehow contribute. Thus, since we don’t expect a simple answer, we feel that this issue is beyond the scope of the present study.

Reviewer #2:[…]1) The authors suggest that the inspiratory phase is regulated by the balance of inhibition, excitation, and intrinsic bursting properties of active neurons, and that in the present experiments with SP, the balance of excitation-inhibition is maintained. While the former is certainly the case in general, excitatory and inhibitory inputs to any neuron in the present experiments have not been directly measured (e.g., by voltage clamp measurements) to know for sure about this balance. Similarly, the authors state, but have not directly shown, that SP shifts the balance of excitation and inhibition among rostral inspiratory neurons, which implies that they have performed some analyses of synaptic interactions among neurons in the network, including in preBötC circuits. This needs to be qualified further in the text. I agree that it is important that the data presented show that SP does not change spiking frequency of excitatory inspiratory neurons during the burst phase as well as activity of inhibitory neurons (and by inference probably local inhibitory interactions) during this phase. Perhaps the modulation of leak conductances by SP (noted in the Introduction) can cause depolarizing shifts in neuronal membrane potentials to promote pre-I spiking without altering overall spiking profiles during the inspiratory burst phase.

It is true that we have not directly assessed the balance of excitatory and inhibitory synaptic input during inspiratory bursts. Indeed, as the reviewer points out, we infer that changes in synaptic interactions reflect the changes in spiking activity we observed under current-clamp conditions. We now consider this limitation throughout the text, and specifically address it in the second paragraph of the Discussion.

However, as the reviewer also points out, it is generally accepted that excitatory and inhibitory preBӧtC neurons are interconnected and, through local synaptic interactions, synchronize together during inspiratory bursts. Indeed, the ability of the isolated preBӧtC network to synchronize depends on local synaptic interactions among excitatory neurons; this is true regardless of which hypothesis of rhythm generation one prescribes to (e.g. “intrinsic bursting”, “group pacemaker”, “burstlet”, or hybrid). Furthermore, in preBӧtC slices (i) inhibitory neurons have inspiratory activity that is dependent on excitatory synaptic transmission (thus, inhibitory neurons receive excitatory inputs during bursts), (ii) the activity of excitatory neurons during bursts is enhanced following blockade of inhibitory synaptic interactions (thus, excitatory neurons receive inhibitory inputs during bursts), and (iii) inspiratory neurons exhibit EPSCs and IPSCs during bursts.

The reviewer’s suggestion that the modulation of leak conductances by SP causes depolarization and promotes an increase spiking during the IBI is exactly what we presume to be true. However, in this study we did not apply substance P following blockade of synaptic transmission to directly assess intrinsic membrane responses to SP and, therefore, we do not want to make any specific claims to correlate membrane depolarization with changes in spiking activity. For more on this, please see our responses to reviewer 1.

2) The baseline membrane potentials of neurons before (and therefore) after SP are not indicated in any of the figures (Figure 2, 3, 4, 5) with whole-cell recordings and provided in the text. This information is important and should be provided to understand why the subpopulation of pre-I excitatory neurons is active during the inter-burst interval (presumably they have more depolarized baseline potentials?), the amount of depolarization if any with SP, and why specific neuron types are recruited or not to spike during the IBI interval. It has been postulated for many years that neurons with more depolarized basal membrane potentials exhibit pre-I spiking activity that is important for rhythmogenesis by providing synaptic excitation during the pre-I spiking phase (e.g., Butera et al., 1999).

We now include V_m_ information for all representative whole-cell recordings. However, a detailed analysis of V_m_ information and discussion regarding SP-induced changes in V_m_ were omitted in the original submission because: (i) Some of the data were obtained in cell-attached configuration; (ii) SP was not applied under conditions with synaptic transmission blocked. Therefore, it cannot be determined whether changes in V_m_ are due to an intrinsic response of the neuron to SP and/or by changes in the synaptic inputs to that neuron; (iii) Vm changes in the cell body may not reflect changes in the dendrites. Thus, the presence or absence of V_m_ changes are difficult to interpret, a challenge that we faced when characterizing the role of the Ih current e.g. (iv) SP-induced changes in V_m_ of preBӧtC neurons have been reported previously using a more appropriate experimental design (e.g. Pena and Ramirez, 2004; Gray et al., 1999), which we have cited in our paper.

We agree with the reviewer that pre-I excitatory neurons likely have a more depolarized basal V_m_, as suggested by previous studies (Smith et al., 2000; Butera et al., 1999), and now include this important point in the Discussion (sixth paragraph).

Reviewer #3:The authors confirm that SP decreases the interval between respiratory bursts and show this shortening is due to shrinking of a later portion of the interval that they dub the "percolation" phase. This point is cleanly and clearly demonstrated in Figure 1, but this contribution alone does not substantially deepen our understanding of respiratory rhythm generation and modulation. Remaining experiments are purely descriptive, flawed with technical issues, and do not provide the proposed mechanistic understanding. Given Yeh et al., 2017, defined the molecular mechanism for SP modulation of breathing, this work does not provide any advancement necessitating publication to a general audience as is.

It is unfortunate that the reviewer did not see much conceptual advance in our study. Although we agree with the reviewer that the study by Yeh et al. was an important contribution to our understanding of SP modulation, we are surprised that the reviewer claims that our intracellular recording experiments are “purely descriptive”, “flawed”, and “do not provide mechanistic understanding”.

Yeh et al., 2017, demonstrated that breathing is destabilized in Nalcn KO mice and that they do not respond to SP. It is a stretch to claim that by describing the consequences of the Nalcn knock-out they defined “the” mechanism for SP modulation, as this description does not explain *how* activation of Nalcn causes changes to breathing. Similarly, our previously published finding that SP modulates I_NaP_ and I_CAN_ is an interesting description, but also does not explain how SP increases frequency. An NK_1_R KO mouse won’t respond to SP either, but should NK_1_R activation be considered “the” mechanism for SP modulation of breathing frequency? While molecular components such as NK_1_R or Nalcn may be required for SP modulation, this does not explain the network changes that actually underlie modulation of breathing frequency and stability. Here, we provide a detailed *network* mechanism, complete with the essential cellular level recordings required to understand network function. We hope the reviewer will agree, that defining a molecular mechanism is only one aspect in unraveling neuromodulation, and that it is essential to integrate all levels in order to fully understand how neural networks change their activity. Otherwise, understanding breathing would have been solved decades ago: as the reviewer knows, TTX blocks breathing which indicates that TTX-sensitive sodium channels are required for breathing generation.

Our cellular and network level analysis strongly support a novel framework for understanding the dynamic regulation of breathing: That there are three inspiratory phases that can be independently modulated to control the frequency and stability of the rhythm. As reviewer 2 pointed out correctly, this will raise many important questions, including follow-up studies to explore how post inspiratory or late expiratory activity influences these three phases in the intact animal. Or more specifically: does the three-phase activity of inspiratory rhythmogenesis serve as a scaffold for the three-phase breathing rhythm? Addressing these questions is conceptually interesting, but beyond the scope of this study. What we know so far is that inhibitory preBӧtC neurons regulate frequency by controlling activity during the burst phase. Here we show that SP specifically regulates frequency by affecting the activity of excitatory neurons during the percolation phase of the rhythm (and we go on to identify the underling network mechanisms that allow the tonic modulatory effects of SP to have phase specific effects on network function). In doing so we provide an important conceptual advance that we feel does in fact “substantially deepen our understanding of respiratory rhythm generation and modulation”.

1) One primary focus of the paper is to record from previously defined populations of pre-inspiratory and inspiratory excitatory neurons and inspiratory inhibitory neurons to determine if SP has cell-type specific effects. However, a major technical flaw throughout the manuscript is the use of current clamp recordings instead of cell attached recordings. Although the authors claim to have done both in the Materials and methods, no cell-attached recordings are shown. In current clamp, the experimenter has complete control of the cells resting membrane potential and can therefore easily make any cell look pre-inspiratory or inspiratory. With this flaw in mind, the authors cannot then claim to have identified cell-type specific modulation by substance P.

Unfortunately, the reviewer has misunderstood our experiments. Thus, we have added more information to the Materials and methods section to clarify.

Current-clamp allows the experimenter to manipulate current to observe changes in voltage. But, this does not mean that it is necessary to apply current to a cell while recording in current clamp mode. In fact, the cell’s true “resting” membrane potential would most certainly not be determined while applying an artificial current. Although many of our recordings were performed in “current-clamp” mode, artificial currents were not applied to control spiking activity or to shift membrane potential from its normal resting value. Moreover, in all experiments, recording settings used to obtain baseline values were maintained constant throughout the experiment – i.e. for the duration of SP application. Thus, changes in spiking induced by SP cannot be attributed to us making “any cell look pre-inspiratory or inspiratory”.

Further, we assume that this reviewer is aware that we are not the first to report that many inspiratory neurons have pre-inspiratory spiking activity (including studies using sharp electrode recording and computational modelling approaches). This has been known for decades, and we assume that the reviewer doesn’t want to imply that all the published electrophysiological studies that characterized the different respiratory discharge patterns using current clamp, have manipulated these patterns at will. In any case, to address the reviewer’s question we now include the requested cell attached recordings, shown in Figure 2—figure supplement 1.

2) In Figure 2E/F and Figure 6 the authors focus attention to the effect of substance P on the onset of neural bursting activity versus preBötC population bursting. They claim that SP increases variability in this timing for pre-inspiratory neurons and that this is key to ensure the preBötC burst size does not increase. As shown in Figure 2E, the onset of the burst appears arbitrarily defined and as shown, would be nearly impossible to definitively decide for a pre-inspiratory neuron. Furthermore, this point is influenced by what membrane potential the experimenter is holding the neuron at in current clamp.

It seems that the reviewer misunderstood our claims. Yes, we observed highly significant changes in the timing variability between the onset of individual neuronal bursts and the onset of integrated activity of the population. This is an issue that we are very interested in, as much can be learned from quantitatively characterizing the striking onset variability of the respiratory network (Carroll and Ramirez, 2013; Carroll et al., 2013). In the Results, we now more specifically define how onset times were determined (subsection “Inspiratory spiking patterns of excitatory and inhibitory neurons in the preBӧtC”, last paragraph). We found that SP increases onset jitter specifically in neurons *without* pre-I activity and does not change onset jitter in pre-I neurons, which is large under baseline conditions. The large onset jitter in pre-I neurons confirms our extracellular recordings as published in Carroll and Ramirez, 2013 – i.e. in the absence of any potential current clamp manipulation. Onset times in pre-I neurons were not “arbitrarily defined”, and regardless, our data indicate, and we suggest, that this subgroup of neurons does not contribute to the increased stochasticity we observed. We *propose as a hypothesis* that this increase in stochasticity among excitatory preBӧtC neurons *may* be one mechanism that prevents these neurons from becoming hyperexcited during inspiratory bursts. However, the observation of burst onset stochasticity is not new and has been previously described not only by us, but also others (e.g. Carroll and Ramirez, 2013; Smith et al., 2000) without using intracellular recordings.

3) Many years of papers have shown that SP increases the preBötC rhythm in a coronal slice preparation. This result shows that brainstem regions rostral to the preBötC are not required for SP modulation of preBötC activity. However, in Figures 7/8 the authors record from neurons rostral to the preBötC and observe changes in activity after SP. They claim these changes cause a shift to increased inhibitory neuron activity that helps to maintain preBötC excitation. This is never directly measured and there are too many logical leaps here, such as:i) Do these rostral neurons project to the preBötC?ii) Are there decreased excitatory inputs into the preBötC?iii) Does bursting in the preBötC become "over-excited" in a coronal slice?iv) If SP acts through Nalcn to depolarize cells, how is it decreasing excitatory neural activity? Given the lack of understanding, these figures add confusion.

Yes, many papers have shown that SP increases frequency in the transverse slice, including from our lab (e.g. Pena and Ramirez, 2002; Telgkamp et al., 2002). We fully agree with the reviewer that this indicates that the rostral area is not required. We are puzzled by the reviewer’s comment, since nowhere in the manuscript do we suggest that rostral areas are *required* for SP modulation of the inspiratory rhythm. However, one cannot automatically assume that if an area is not required, that it does not contribute to a certain phenomenon. To emphasize this logic, here is one example: locomotor and breathing activity do not require the neocortex, but certainly, the neocortex plays an important role in these behaviors. In fact, our recordings from rostral inspiratory neurons showed relatively heterogenous responses to SP, which was explicitly stated in our original submission. This is an important result, however, because it suggests not only that the rostral area is not required, but also that any contributions from rostral inspiratory neurons to SP-induced frequency facilitation are relatively modest. This new insight could not be revealed using transverse slice preparations. In the revised text, we propose that the modest contribution of the rostral area may be due to the enriched expression of NK_1_R in the preBӧtC relative to the rest of the VRC. Indeed, NK_1_R is a classical “marker” of the preBӧtC (Gray et al., 1999).

Since we do not suggest that changes rostral are a significant contributor to SP-induced modulation of breathing, we feel that it is unnecessary to address the “logical leaps” implied by the reviewer.

[Editors’ note: the author responses to the re-review follow.]

The authors have done an admirable job responding to the previous review. One reviewer ask for further clarifications, which should be seriously considered. When the authors re-submit no external further review will be necessary.Essential revisions:The revised manuscript has nicely incorporated the feedback from the three essential revisions. The inclusion of the membrane potential in the featured recordings as well as the number of cell-attached recordings performed addresses essential revision comment #2. Although the N is low for the number of cell-attached recordings, this data is important to confirm that a subset of excitatory-inspiratory neurons display pre-I activity only after SP application. Additionally, as the authors point out, the described membrane potentials suggest that the recruited neurons are likely those with higher resting membrane potentials before SP application. With this in mind, the authors should consider providing the membrane potential before and after SP for all of the recorded neurons of the various neural types. While the depolarization and recruitment of cells into pre-I firing is interesting, it will be essential in the future to determine if the recruitment is required for the SP induced shortening of the percolation phase. Alternative models still remain, like the increased activity of pre-I neurons alone drives the shortening of the percolation phase. The additional description of the methods will enhance reproducibility by others and the modified Discussion nicely handles how the various observations made in the manuscript may result in the observed selective effect of SP to shorten the percolation phase and addresses essential revision comments #1 and #3.

We appreciate the reviewers for taking the time to carefully assess our study and thank them for their helpful critiques.

We now include quantified V_m_ information for whole-cell recordings. We did see a small increase in V_m_ of pre-I neurons in response to SP, and there was also a trend (p=0.07) towards an increase in V_m_ among excitatory neurons that were recruited to pre-I. However, we refrain from detailed discussion of this in the text since the cell attached recordings cannot be included and because we did not test whether changes in V_m_ are due to an intrinsic response of the neuron to SP and/or by changes in the synaptic inputs to that neuron.

We agree with the reviewer that it will be important for future studies to more directly test whether the recruitment of excitatory neurons into pre-I spiking is necessary for SP induced facilitation. However, we did find that the slope of pre-I spiking in both pre-I neurons and neurons recruited to pre-I can predict the duration between bursts, strongly suggesting that these processes are linked.